# JudgeAgent: Knowledge-wise and Dynamic LLM Evaluation with Agent-as-Interviewer

## Abstract

Current evaluation paradigms for large language models (LLMs) suffer from over-estimated or biased evaluation and mismatched question difficulty, leading to incomplete evaluations of LLM's knowledge and capability boundaries, which hinder LLM's effective application and optimization. To address these challenges, we propose Agent-as-Interviewer, a dynamic evaluation paradigm that employs LLM agents to conduct multi-turn interactions for evaluation. Unlike current benchmarking or dynamic interaction paradigms, Agent-as-Interviewer utilizes agents to call knowledge tools for wider and deeper knowledge in the dynamic multi-turn question generation, achieving more complete evaluations of the LLM's knowledge boundaries. It also leverages agents to plan query strategies for adjustment of the question difficulty levels, enhancing the difficulty control to match the actual capabilities of target LLMs. Based on this paradigm, we develop JudgeAgent, a knowledge-wise dynamic evaluation framework that employs knowledge-driven synthesis as the agent's tool, and uses difficulty scoring as strategy guidance, thereby finally providing valuable suggestions to help targets optimize themselves. Extensive experiments validate the effectiveness of JudgeAgent's suggestions, demonstrating that Agent-as-Interviewer can accurately identify the knowledge and capability boundaries of target models. The source code is available on
https://anonymous.4open.science/r/JudgeAgent.

## 1 Introduction

Evaluating large language models (LLMs) to understand the boundaries of their knowledge and capabilities is a critical step for their successful application across various domains (Tang et al., 2024; Yuan et al., 2023; Shi, 2024; Wang et al., 2025). Current mainstream evaluation methods for LLMs rely on static benchmark evaluations (Clark et al., 2018; Hendrycks et al., 2021; Huang et al., 2023; Lin et al., 2022; Cobbe et al., 2021; Tang & Yang, 2024), where predefined questions are posed to the target LLM and the target's performance is evaluated by humans (Chang et al., 2024) or LLM-as-a-judge (Liu et al., 2023; Wang et al., 2023b; Zheng et al., 2023). Benefiting from the controlled question quality and straightforward workflows of the static benchmarking paradigm, developers can rapidly acquire a fundamental understanding of the strengths and limitations of various evolving LLMs, thereby facilitating swift iteration in LLM applications.

However, this benchmarking paradigm faces the bottleneck that benchmarks have become saturated more and more quickly in recent years. It took until 2023 for the accuracy record of MMLU(Hendrycks et al., 2020), which was released in 2020, to reach 80%. While the GPQA benchmark(Rein et al., 2023), published in late 2023, achieved an 80% record just within one year. The bottleneck is due to the static nature(Gu et al., 2024; Ko et al., 2024) of this paradigm. Firstly, the static paradigm restricts evaluations within a predefined knowledge scope, making it difficult to accurately measure the model's comprehensive knowledge mastery (Wang et al., 2023a; Kwan et al., 2024), leading to out-of-domain risks in practical applications. Secondly, the lack of dynamic updates increases the risk of data contamination(Schaeffer, 2023; Oren et al., 2023), where target models artificially inflate their benchmark performance by being exposed to test questions during training, leading to an overestimation of their actual capabilities. These shortcomings result in LLMs being misapplied in scenarios beyond their actual capabilities, causing the waste of resources.

Figure 1: The difference between Agent-as-Interviewer and current evaluation paradigms.

Consequently, researchers begin to focus on dynamic evaluation paradigms that leverage LLMs to dynamically modify questions. Initial efforts involve using LLMs to modify questions from static benchmarks based on specific requirements to mildly mitigate data contamination (Bai et al., 2023; Shi et al., 2025). Beyond these methods, researchers proposed paradigms that dynamically adjust questions during multi-turn interactions based on the model's previous responses, enabling a more in-depth evaluation (Wang et al., 2023a; Kim et al., 2025). While these dynamic approaches partially address challenges in static paradigms, such as data contamination and plain evaluations, they still suffer from biased evaluation and mismatched difficulty due to simplified feedback mechanisms and knowledge limitations of evaluator LLMs, leading to a misalignment between evaluation questions and the actual range of the target's capability. This misalignment hinders precise evaluations of the target LLM's knowledge and capability boundaries, thereby impeding effective guidance for both application and subsequent optimization of LLMs.

To address these challenges, we propose Agent-as-Interviewer, a dynamic evaluation paradigm that employs LLM agents to conduct multi-turn interactions for evaluation. Unlike current benchmarking or dynamic interaction paradigms, in the dynamic follow-up question generation based on the target's responses in multi-turn interactions, Agent-as-Interviewer utilizes agents to call knowledge tools for wider and deeper knowledge, achieving more complete evaluations of the target LLM's knowledge and capability boundaries. It also leverages agents to plan query strategies for adjustment of the question difficulty levels, enhancing the difficulty control to match the actual capabilities of target LLMs. Based on this paradigm, we develop JudgeAgent, a knowledge-wise dynamic evaluation framework that employs knowledge-driven synthesis as the agent's tool and uses difficulty scoring as strategy guidance in multi-turn dynamic interactions, thereby finally providing valuable suggestions to help targets optimize themselves.

In summary, our contributions are as follows:

- **Agent-as-Interviewer paradigm**: We propose a dynamic evaluation paradigm that employs LLM agents to conduct multi-turn interactions for evaluation, addressing the challenges of incomplete evaluation and inadequate difficulty control in current paradigms.

- **Knowledge-wise evaluation framework**: We introduce JudgeAgent, a knowledge-wise dynamic evaluation framework based on Agent-as-Interviewer, to provide valuable suggestions and precise guidance for the optimization of target LLMs.

- **Validation of the paradigm**: We conduct thorough experiments and analysis to validate the effectiveness of Agent-as-Interviewer and JudgeAgent's suggestions.

## 2 RELATED WORKS

### 2.1 STATIC BENCHMARK-BASED EVALUATION

These methods employ pre-constructed benchmarks to evaluate LLMs in specific tasks, using formats such as multiple-choice, question-answer(Q&A), or prompts for performing tasks. For multiple-choice (Clark et al., 2018; Hendrycks et al., 2021; Huang et al., 2023) or Q&A (Lin et al., 2022; Cobbe et al., 2021; Tang & Yang, 2024) formats, the benchmark provides correct answers and measures the target's capability by its accuracy. For task-execution format, the benchmark measures

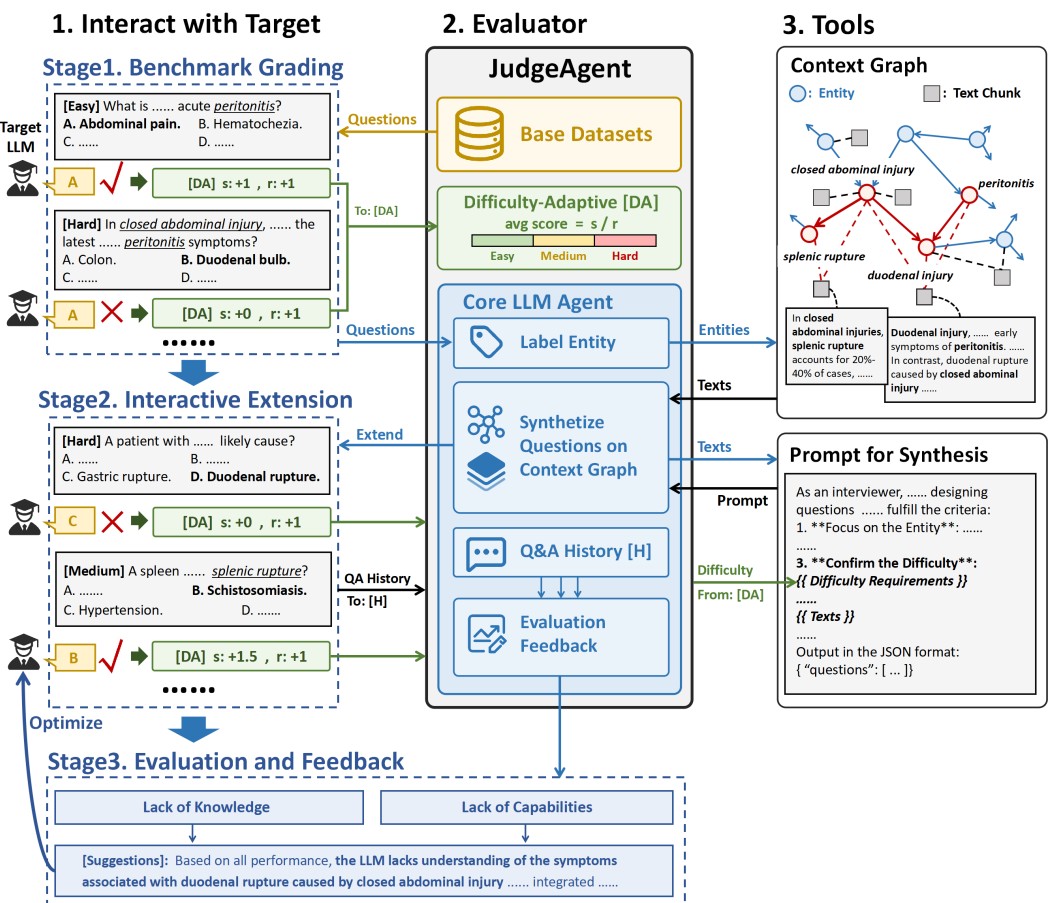

Figure 2: The framework of JudgeAgent. The left part is the interaction process. The central part is the composition of JudgeAgent. The right part presents the tools of JudgeAgent.

the model by the correctness and plausibility assessed by human (Chang et al., 2024) or LLM-as-a-judge (Wang et al., 2023b; Zheng et al., 2023) based on reference answers. These methods ensure controllable question quality and rapid pipelines, but they are also susceptible to data contamination, which undermines the validity of evaluations.

## 2.2 DYNAMIC LLM-BASED EVALUATION

These methods leverage LLMs to dynamically generate evaluation questions. Bai et al. (2023) employs LLMs as examiners to generate questions based on Google Trends categories. SafetyQuizzer(Shi et al., 2025) formulates questions with current events retrieved from search engines to maximize the timeliness. These methods focus on the knowledge sources, but lack interaction with target models. Therefore, researchers attempt to generate follow-up questions based on responses in multi-round Q&A sessions. KIEval (Yu et al., 2024) and LLM-as-an-Interviewer (Kim et al., 2025) leverage LLMs to generate follow-up questions and provide feedback based on the target's responses in multi-turn interactions, thereby probing the depth of the target's knowledge.

Compared to static methods, their advantage is that target models cannot prepare for generated questions, thereby producing more genuine evaluations. However, these dynamic processes are often centered on a single question, and all the generations rely entirely on the LLM, resulting in a limited scope of knowledge coverage and pool quality control. Moreover, these methods focus on the content of questions, without properly calibrating question difficulty to the target's capability range, leading to biased evaluations. To address these challenges, we propose Agent-as-Interviewer and develop JudgeAgent based on this paradigm, which performs fine-grained evaluation through knowledge-wise generation and adaptive difficulty adjustment in dynamic multi-turn interactions.

## 3 METHODOLOGY

We introduce Agent-as-Interviewer, which utilizes LLM agents to simulate the entire interview for the dynamic evaluation of LLMs, and develop JudgeAgent based on this paradigm. In this section, we use JudgeAgent as an example to illustrate the workflow of this paradigm. The workflow comprises three core components: **(1) Benchmark Grading**: Fundamentally estimating the target's capability range through testing on public static benchmarks. **(2) Interactive Extension**: Using knowledge-driven data synthesis to dynamically generate follow-up questions and update the difficulty levels based on the target's capability estimations. **(3) Evaluation Feedback**: Evaluating the target's deficiencies in multiple dimensions and providing actionable suggestions for optimization. The overall framework is illustrated in Figure 2, and detailed as pseudo code in Algorithm 1.

### 3.1 BENCHMARK GRADING

Analogous to a written test before an interview, JudgeAgent forms a base understanding of the target model's capabilities by evaluating its performance on a publicly available static benchmark.

JudgeAgent first partitions the benchmark questions into batches and then prompts the target to answer the questions batch by batch. After assessing the target's answers, JudgeAgent estimates the target's capability based on its performance in each batch. Within each batch, each correct answer improves the total score, and the average score determines the follow-up question's difficulty in the extension stage, including *Easy*, *Medium*, and *Hard*. Let $[d_1, d_2, d_3]$ respectively denote the difficulty *Easy*, *Medium*, and *Hard*, $n_{it}, n_{if}$ respectively represent the number of correct and incorrect answers for $d_i$-level questions, $c_{it}$ be the score gain for a correct answer at $d_i$, and $t_{ij}$ be the score threshold between $d_i$ and $d_j$. The difficulty control mechanism is formalized as follows:

$$avg_s = \frac{\sum_{i=1}^{3} c_{it} n_{it}}{\sum_{i=1}^{3} (n_{it} + n_{if})} \quad , \quad d_{next} = \begin{cases} d_1, & avg_s \leq t_{12} \\ d_2, & avg_s \leq t_{23} \\ d_3, & avg_s > t_{23} \end{cases} \quad (1)$$

In JudgeAgent, we set $[c_{1t}, c_{2t}, c_{3t}] = [1, 1.5, 2]$ and $[t_{12}, t_{23}] = [0.5, 1]$. For benchmarks without difficulty labels, the default score gain is $c_{2t}$. The mechanism's detailed design rules, analysis, and proofs are available in Appendix B.

### 3.2 INTERACTIVE EXTENSION

This process is an "interview" where JudgeAgent conducts an in-depth evaluation of target models. Specifically, JudgeAgent iteratively generates questions based on the target's responses and dynamically controls the difficulty. The process consists of three steps: Relevant Knowledge Retrieval, Difficulty-Adaptive Question Generation, and Capability Estimation.

**Relevant Knowledge Retrieval:** Based on seed question batches, JudgeAgent retrieves related knowledge from the benchmark's knowledge base. If only reference texts of seed questions are used as background knowledge for generation, the generated questions will be highly similar to the original questions, with limited knowledge scopes. Inspired by SoG(Jiang et al., 2025), JudgeAgent employs a context-graph–based sampling approach to get background texts. The context graph is constructed from the benchmark's knowledge base (e.g., the collection of all reference texts) by segmenting texts into chunks and extracting entities as nodes, with entities in the same text as neighbors. The detailed construction is illustrated in Appendix C and Algorithm 3.

During sampling, JudgeAgent first extracts entities from seed questions and finds the most similar entity $e$ on the graph. In each hop of sampling, JudgeAgent retrieves the most relevant chunks of $e$ to the question, randomly selects a chunk $c$, treats $(e, c)$ as the root, and $e$ is used as the entity for the next hop. This process is repeated for $N$ hops to form a knowledge path with breadth and depth, and the chunks along the path are concatenated to form the background knowledge. The context graph ensures both knowledge breadth and depth, and the greedy similarity strategy ensures relevance to seed questions. Entity extraction is performed by an LLM, which is GPT-4.1 in our experiments, with prompts detailed in Appendix G. This step provides the appropriate knowledge background for JudgeAgent to generate knowledge-wise questions.

**Difficulty-Adaptive Question Generation:** Unlike other dynamic methods, JudgeAgent incorporates the target's capability estimation into question generation to enable more fine-grained evaluations. A fixed-difficulty test can identify underperforming models, but can hardly comprehensively evaluate a model's knowledge levels. For instance, when an overly challenging test evaluates a model with limited knowledge, the evaluation may reveal that the model performs poorly, but fails to delineate the actual range of its knowledge. Therefore, JudgeAgent dynamically adjusts the target's capability estimation and the follow-up question's difficulty based on Eq 1. This adaptive approach allows for a precise and refined understanding of the target's knowledge boundaries.

Based on the capability estimation, JudgeAgent adjusts the question difficulty as *Easy*, *Medium*, or *Hard*, and explicitly specifies each difficulty's requirements in the prompt. For *Easy*, the focus is on assessing information retrieval skills, primarily through cloze-style questions based on the original text, like *"Which virus primarily causes HFMD?"*. For *Medium*, the emphasis is on understanding key concepts, with questions involving basic inference, like *"Which description is correct regarding meiosis II?"* For *Hard*, the questions are designed to encourage deep thinking and complex logical analysis, like *"A patient ingested a toxic substance... The doctor employed... which type of treatment is likely lacking?"*. Detailed prompt designs are provided in Appendix G.

**Capability Estimation:** In each round, JudgeAgent presents the generated questions to the target model and collects its responses. Based on all responses in the above two stages, JudgeAgent computes scores according to the scoring scheme described in Section 3.1, considering both difficulty and correctness. The average score is used to determine the difficulty for the next round's generation.

### 3.3 EVALUATION FEEDBACK

This process constitutes the result generation phase. JudgeAgent aggregates the target's all Q&A performances in the above two stages, and evaluates the target from multiple dimensions, including knowledge boundaries, logical reasoning, and comprehensive performance, producing a text evaluation report. Based on this evaluation, JudgeAgent simultaneously provides suggestions aimed at addressing the identified capability gaps, serving as a reference for model optimization.

Inspired by Generative Reward Model methods (Ankner et al., 2024; Ye et al., 2024; Li et al., 2025) and the evaluation report method proposed by (Kim et al., 2025), JudgeAgent generates feedback and suggestions based on all Q&A history for each batch, without directly providing the correct answer or background knowledge. Benefiting from context graphs, the suggestions can identify key knowledge concepts and extend relevant information to provide concise and valuable feedback, rather than providing "cheating information" specific to seed questions. To verify the suggestion's effectiveness, the target model is prompted to answer the same questions again with these suggestions. By comparing accuracy before and after the intervention, we can indirectly validate the effectiveness of the evaluations.

## 4 EXPERIMENTS

In this section, we validate the effectiveness of JudgeAgent's evaluations using the method in Section 3.3, which is detailed in Algorithm 2. The following research questions guide our experiments: First, does JudgeAgent genuinely discover the shortcomings of the target model (**RQ1**)? Second, to what extent does each mechanism in JudgeAgent influence the evaluations (**RQ2**)?

**Experiment Setup.** We select GPT-4.1[1](Achiam et al., 2023) to be the core LLM in JudgeAgent, which serves as both a generator and evaluator model. In our experiments, the batch size in Benchmark Grading is 3, and the Interactive Extension is limited to a maximum of 3 rounds.

### 4.1 DATASET AND TARGET MODEL SELECTION

The initial stage of JudgeAgent, Benchmark Grading, requires a predefined static benchmark. In our experiments, we select MedQA(Jin et al., 2021), MultiHop-RAG(Tang & Yang, 2024), and QuALITY(Pang et al., 2022). We remove the background knowledge of questions from MedQA and MultiHop-RAG during evaluation to evaluate the target's knowledge boundaries rather than

---

[1]We use gpt-4.1-2025-04-14 from OpenAI's official API for all experiments

its comprehension ability, and to verify whether JudgeAgent can discover the target's knowledge deficiencies. We use QuALITY to validate the effectiveness of JudgeAgent in guiding the comprehension and reasoning of target models. For detailed statistical information and the preprocessing procedures of these datasets in our experiments, refer to Appendix D.

For the target models, we select three LLMs as the targets: Qwen3(Yang et al., 2025), GLM4-Flash(GLM et al., 2024), GPT-4.1(Achiam et al., 2023), and gemini-2.5-pro(Comanici et al., 2025)[2]. We utilize the official APIs to interact with these models.

## 4.2 EVALUATION METRICS

The metrics used in our experiments are as follows:

(1) **Accuracy (ACC).** We use this metric to measure the performance of the target model on base datasets, which is the proportion of questions correctly answered by the target model. To investigate the validity of JudgeAgent's evaluations, we compared the changes in the ACC before and after dynamic evaluation. ACC1 represents the target model's ACC in answering base questions in *Benchmark Grading* stage, measuring its performance before receiving evaluation feedback. ACC2 denotes the ACC in answering the same questions after receiving feedback from JudgeAgent.

(2) **Correction Rate (CR).** This metric quantifies the proportion of questions that the target model initially answered incorrectly but subsequently answered correctly after receiving evaluation suggestions, which measures the effectiveness of the feedback suggestions from JudgeAgent. A higher correction rate indicates better performance of the suggestions.

(3) **Correct-to-Error Rate (CtE).** This metric serves as the inverse of the Correction Rate, representing the proportion of questions that were initially answered correctly but subsequently answered incorrectly. A lower CtE indicates greater effectiveness of the suggestions.

## 4.3 MAIN RESULTS AND ANALYSIS

To address **RQ1**, we conduct experiments on three datasets to validate the effectiveness of JudgeAgent's evaluation, and the results are shown in Table 1 and Table 2.

Based on the results from the knowledge-intensive datasets, MedQA and MultiHopRAG, as presented in Table 1, JudgeAgent can effectively identify potential knowledge gaps in the target LLM and subsequently mitigate these gaps by providing targeted prompts that indicate possibly missing knowledge points or overlooked knowledge associations to the target models. Furthermore, as evidenced by the overall performance on QuALITY, a dataset emphasizing reasoning and comprehension, as presented in Table 2, JudgeAgent also contributes to providing further optimization guidance to address potential shortcomings in the logical reasoning and semantic understanding abilities of the target models, thereby assisting in refining the target's thinking steps. Additionally, by comparing the Correction Rate (CR) and the Correction-to-Error Rate (CtE) of different models before and after receiving evaluation suggestions, it can be observed that the effectiveness of the suggestions is less consistent for relatively weaker models (e.g., the free model GLM4-Flash), as reflected in the higher CtE. In contrast, stronger models are less susceptible to misleading suggestions.

Based on the performance across different difficulties of QuALITY as shown in Table 2, we can analyze the effectiveness of JudgeAgent's suggestions for different targets on various difficulties. For stronger models (Qwen3, GPT-4.1, and Gemini-2.5-pro), JudgeAgent provides stronger guidance and optimization, particularly on questions of higher difficulty (*Medium* and *Hard*), since these models may have already mastered the basic knowledge of *Easy*-level questions. In contrast, for the weaker models, JudgeAgent leads to considerable improvement across all difficulty levels, with particularly notable gains on the *Easy* level, since JudgeAgent's feedback is more effective at filling gaps in basic concepts. These results further indicate that JudgeAgent more accurately assesses the capability boundaries of targets and provides more difficulty-adaptive optimization to the targets. Furthermore, for all target models, JudgeAgent's suggestions demonstrate clear optimizing guidance on *Hard*-level questions that involve complex reasoning, suggesting that JudgeAgent effectively identifies underlying deficiencies in target models through dynamic interactive evaluation.

---

[2]We use qwen-plus-2025-04-28, gpt-4.1-2025-04-14, gemini-2.5-pro-preview-06-05, and the free version glm-4-flash-250414 as the target models.

Table 1: The results on MedQA and MultiHopRAG, and all values are percentages.

| Target Model | MedQA | | | | MultiHopRAG | | | |
|---|---|---|---|---|---|---|---|---|
| | ACC1 | ACC2 | CR↑ | CtE↓ | ACC1 | ACC2 | CR↑ | CtE↓ |
| Qwen3 | 91.71 | 96.38 | 5.02 | 0.35 | 63.65 | 70.07 | 17.49 | 11.07 |
| GLM4-Flash | 80.09 | 92.82 | 13.46 | 0.73 | 51.25 | 65.92 | 24.26 | 9.59 |
| GPT-4.1 | 84.97 | 92.44 | 7.65 | 0.18 | 68.94 | 75.55 | 12.36 | 5.75 |
| Gemini-2.5-pro | 91.04 | 94.60 | 4.03 | 0.47 | 62.25 | 71.83 | 15.41 | 5.83 |

Table 2: The results on QuALITY with different difficulty levels. QuALITY-X refers to the sub-dataset that consists of questions with specific difficulty, and -overall refers to all questions.

| Target Model | QuALITY-overall | | | | QuALITY-easy | | | |
|---|---|---|---|---|---|---|---|---|
| | ACC1 | ACC2 | CR↑ | CtE↓ | ACC1 | ACC2 | CR↑ | CtE↓ |
| Qwen3 | 87.83 | 88.84 | 1.88 | 0.87 | 94.63 | 95.61 | 0.98 | 0.00 |
| GLM4-Flash | 73.48 | 76.38 | 4.78 | 1.88 | 83.26 | 84.65 | 4.19 | 2.79 |
| GPT-4.1 | 89.13 | 92.75 | 3.91 | 0.29 | 93.66 | 96.10 | 2.44 | 0.00 |
| Gemini-2.5-pro | 93.77 | 96.38 | 3.19 | 0.58 | 97.55 | 99.02 | 1.47 | 0.00 |

| Target Model | QuALITY-medium | | | | QuALITY-hard | | | |
|---|---|---|---|---|---|---|---|---|
| | ACC1 | ACC2 | CR↑ | CtE↓ | ACC1 | ACC2 | CR↑ | CtE↓ |
| Qwen3 | 88.53 | 88.89 | 1.43 | 1.08 | 80.10 | 82.04 | 3.40 | 1.46 |
| GLM4-Flash | 74.35 | 79.18 | 5.58 | 0.74 | 62.14 | 64.08 | 4.37 | 2.43 |
| GPT-4.1 | 91.04 | 94.27 | 3.58 | 0.36 | 82.04 | 87.38 | 5.83 | 0.49 |
| Gemini-2.5-pro | 93.57 | 95.36 | 2.50 | 0.71 | 90.29 | 95.15 | 5.83 | 0.97 |

In summary, the results indicate that JudgeAgent can effectively identify potential knowledge or capability gaps in target models and optimize their performance by providing targeted suggestions. Furthermore, through a difficulty-adaptive, dynamic interaction-based evaluation, JudgeAgent can more precisely delineate the target model's capabilities, offering more refined evaluation results.

In addition, we conducted cross-validation experiments to verify that the JudgeAgent's suggestions are effective not only for base questions, further corroborating the validity of the main experiment results. The results and analysis are detailed in Appendix E.2

## 4.4 ABLATION STUDY

To address **RQ2**, we conduct ablation studies on MedQA with GLM4-Flash as the target LLM. When maintaining the responses to base questions unchanged, we removed different modules of JudgeAgnet. By comparing the performance improvement under various ablation settings, we investigated how much different modules influence the evaluations. The results are shown in Table 3.

JudgeAgent (*w/o* context graph) removes the context graph, which ensures the knowledge relevance between synthesized questions and the base ones, and only uses chunks sampled randomly from the original knowledge base. The results indicate that removing the context graph reduces the improvement effect of the JudgeAgent's evaluations. A possible explanation is that the lack of context graph prevents extended questions and base questions from being associated in terms of focused entities and knowledge, leading to fragmented suggestions without accurate information, which may disrupt the thinking of target LLMs. The higher correction-to-error rate (CtE) compared to the setting -*w/o* difficulty-adaptive also provides supporting evidence for this potential interference.

JudgeAgent (*w/o* difficulty-adaptive) removes the difficulty-adaptive mechanism, which enables JudgeAgent to assess the target model's capability precisely, and generates questions with fixed difficulty rules in the prompt. The results show that the removal of this module diminishes the effectiveness of JudgeAgent's evaluations, demonstrating the importance of the difficulty-adaptive mechanism for providing effective evaluations.

Table 3: The results of the ablation study with MedQA as the base dataset, and all values are percentages. $\Delta = \text{ACC2} - \text{ACC1} = \text{CR} - \text{CtE}$ refers to the overall improvement on the target models after receiving evaluation from JudgeAgent.

| Target Model | Evaluator | ACC1 | MedQA ACC2 | CR↑ | CtE↓ | Δ↑ |
|---|---|---|---|---|---|---|
| GLM4-Flash | JudgeAgent | 80.09 | 92.82 | 13.46 | 0.73 | 12.73 |
| | -*w/o* context graph | | 88.60 | 10.26 | 1.75 | 8.50 |
| | -*w/o* difficulty-adaptive | | 89.68 | 11.14 | 1.56 | 9.59 |
| | -*w/o* interactive extension | | 86.50 | 9.47 | 3.06 | 6.41 |
| GPT-4.1 | JudgeAgent | 84.97 | 92.44 | 7.65 | 0.18 | 7.47 |
| | -*w/o* context graph | | 91.36 | 6.72 | 0.33 | 6.39 |
| | -*w/o* difficulty-adaptive | | 91.94 | 7.36 | 0.38 | 6.98 |
| | -*w/o* interactive extension | | 89.81 | 6.13 | 1.28 | 4.85 |
| Qwen3 | JudgeAgent | 91.71 | 96.38 | 5.02 | 0.35 | 4.67 |
| | -*w/o* context graph | | 95.57 | 4.28 | 0.42 | 3.86 |
| | -*w/o* difficulty-adaptive | | 95.94 | 4.59 | 0.36 | 4.23 |
| | -*w/o* interactive extension | | 93.01 | 1.97 | 0.67 | 1.30 |

JudgeAgent (*w/o* interactive extension) removes the Interactive Extension, which is the core process of dynamic evaluation to extend the evaluation scope of knowledge and capability, and only evaluates models with base datasets. The results show that removing the Interactive Extension significantly weakens the effectiveness of JudgeAgent compared to other ablation settings. This indicates that the evaluation enabled by the dynamic expansion, which expands both breadth and depth of knowledge and capability in dynamic evaluation, is crucial to the final evaluation suggestions.

Additionally, the ablation experiment results comparing different models reveal that for relatively weaker models (e.g., GLM4-Flash), the performance decline after removing each component is more pronounced, which indicates that these dynamic evaluation mechanisms are particularly beneficial to weaker models and can effectively facilitate their optimization.

## 4.5 PARAMETER ANALYSIS

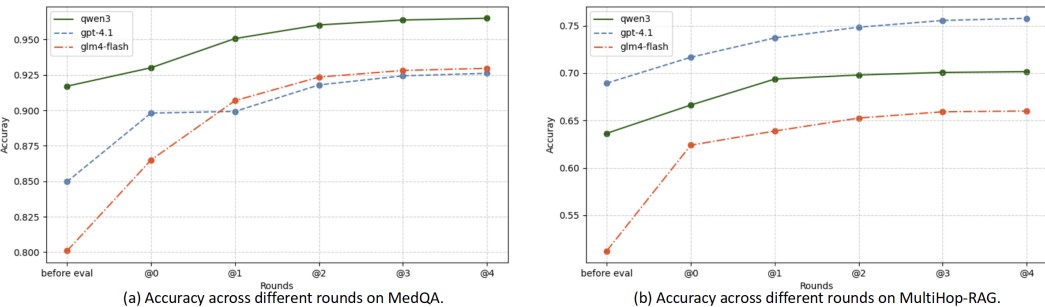

(a) Accuracy across different rounds on MedQA.    (b) Accuracy across different rounds on MultiHop-RAG.

Figure 3: The results of different expansion rounds on MedQA and MultiHop-RAG. @K indicates the ACC improvement after receiving suggestions from the K-th interaction.

**Are more expansion rounds better for JudgeAgent?** To answer this research question, we tested the evaluation effectiveness of JudgeAgent under different rounds in the Interactive Extension, as shown in Figure 3. The results demonstrate that as the expansion rounds increase, the effectiveness of JudgeAgent, which is represented by the accuracy improvement of target models, also gradually improves. However, the trend gradually slows down, showing a relatively clear marginal effect.

As the rounds increase, JudgeAgent can expand more questions around the knowledge related to base questions. After reaching certain rounds, the Q&A history is sufficient for JudgeAgent to identify the target's knowledge deficiencies around base questions. Additional questions serve only as corroborative rather than critical evidence. There is a clear marginal effect in Figure 3. For

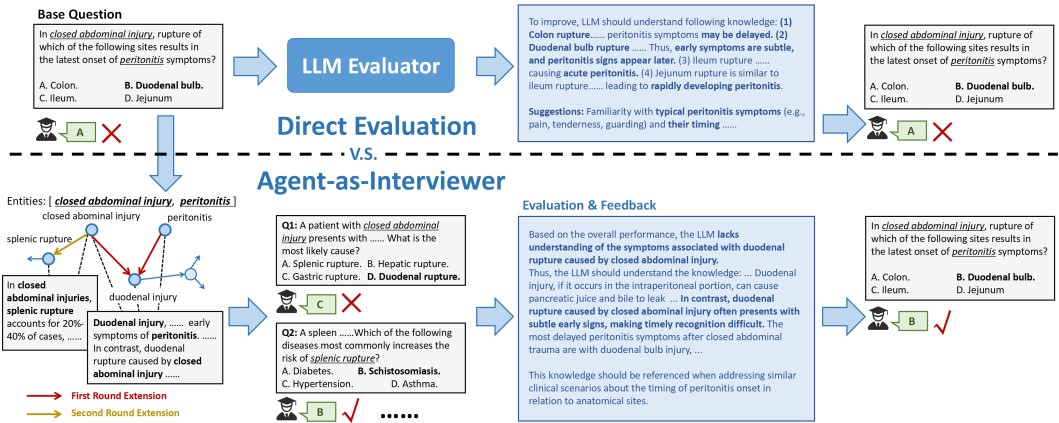

Figure 4: The brief overview of the comparative case in the Case Study.

example, the results of GPT-4.1 on MultiHop-RAG, the improvement per round decreases gradually from 2.04% by the first round, 1.13% by the second, 0.71% by the third, and to 0.23% by the last round, and other curves exhibit similar patterns. Therefore, JudgeAgent expands a maximum of 3 rounds in our experiments to avoid wasting resources and time.

## 4.6 CASE STUDY

To gain a deeper understanding of Agent-as-Interviewer's mechanism, we analyze GLM4-Flash's responses to MedQA questions by receiving suggestions from direct evaluation and Agent-as-Interviewer. The seed questions where the target answers correctly are omitted in this case. A brief overview of the case is shown in Figure 4, while the detailed content is provided in Appendix H.

In this case, the target LLM answers the base question incorrectly. When directly evaluated by the LLM evaluator, the evaluator adopted a more conservative approach to generically enumerate the potential consequences of closed abdominal injury, rather than to perform a comparison between the specific medical concepts involved. Consequently, even after receiving the feedback, the target LLM still outputs an incorrect answer, which is attributable to a nuance it failed to discern: peritonitis symptoms resulting from both colon injury and duodenal bulb injury can exhibit delayed onset.

In contrast, JudgeAgent first extracted two key entities, *closed abdominal injury* and *peritonitis*. It then retrieved relevant entities with texts, such as *splenic rupture* and *duodenal injury*, from the context graph. Based on the sampled knowledge paths, JudgeAgent generated a series of extended questions. Given that the target model consistently errs on all questions where *duodenal injury* and *closed abdominal injury* co-occur, JudgeAgent can determine that the target model had insufficient understanding of this concept entity, thereby providing specific knowledge guidance. Ultimately, the target LLM successfully answered the original base question correctly with this feedback.

The case above demonstrates that through the Agent-as-Interviewer paradigm, JudgeAgent is capable of accurately identifying gaps in the target's knowledge and providing targeted feedback.

## 5 CONCLUSION

In this paper, we propose Agent-as-Interviewer, a dynamic evaluation paradigm that employs LLM agents to conduct multi-turn interactions for evaluation. This paradigm utilizes agents to call knowledge tools for wider and deeper knowledge in the dynamic question generation to achieve more complete evaluations, and leverages agents to plan query strategies for adjustment of the question difficulty to match the actual capabilities of target LLMs. Based on this paradigm, we develop JudgeAgent, a knowledge-wise dynamic evaluation framework to assess LLMs in multi-turn Q&A sessions and provide valuable suggestions that assist target LLMs in optimizing themselves. Thorough experiments validate that Agent-as-Interviewer can precisely assess the target's knowledge boundaries. In our future work, we will further refine this novel evaluation paradigm and develop more reliable evaluation frameworks for LLMs.

ETHICS STATEMENT

This work adheres to the ICLR Code of Ethics. In this study, all datasets used, including MedQA, MultiHop-RAG, and QuALITY, were sourced in compliance with relevant usage guidelines, ensuring no violation of privacy. No personally identifiable information was used, and no experiments were conducted that could raise privacy or security concerns. We are committed to maintaining transparency and integrity throughout the research process.

REPRODUCIBILITY STATEMENT

We have made every effort to ensure that the pipelines and results in this paper are reproducible. All code has been made publicly available in an anonymous repository to facilitate replication and verification, and the URL is at the end of the abstract (`https://anonymous.4open.science/r/JudgeAgent`). The experimental setup is described in detail in Section 4.1 and 4.2. The processing of data is shown in Appendix D. The pipelines of JudgeAgent are detailed as pseudo code in Appendix F. The prompts of JudgeAgent are detailed in Appendix G.

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

## A  THE USE OF LARGE LANGUAGE MODELS

In this paper, we only used LLMs, including DeepSeek-R1 and ChatGPT, for polishing the writing. Specifically, we used LLMs to assist in refining the language, improving readability, and grammar checking. The authors take full responsibility for the content of the paper.

## B  ANALYSIS AND PROOFS OF ADAPTIVE DIFFICULTY-CONTROL MECHANISMS

In Section 3.1, we have formalized the difficulty control mechanisms as follows:

Let $[d_1, d_2, d_3]$ respectively denote the difficulty *Easy*, *Medium*, and *Hard*, $n_{it}, n_{if}$ respectively represent the number of correct and incorrect answers for $d_i$-level questions, $c_{it}$ be the score gain for a correct answer at $d_i$, and $t_{ij}$ be the score threshold between $d_i$ and $d_j$, we can get the average score $avg_s$ and the difficulty of the next round question generation $d_{next}$:

$$avg_s = \frac{\sum_{i=1}^{3} c_{it} n_{it}}{\sum_{i=1}^{3} (n_{it} + n_{if})} \quad , \quad d_{next} = \begin{cases} d_1, & avg_s \leq t_{12} \\ d_2, & avg_s \leq t_{23} \\ d_3, & avg_s > t_{23} \end{cases}$$

where $[c_{1t}, c_{2t}, c_{3t}] = [1, 1.5, 2]$ and $[t_{12}, t_{23}] = [0.5, 1]$ in our experiments. In this section, we will illustrate the criteria and rules of the score gain $c_{it}$ and the threshold $t_{ij}$, along with the analysis and proofs. For ease of representation, let $N = \sum_{i=1}^{3} (n_{it} + n_{if})$ in the subsequent analysis.

For the selection of score gains and thresholds, we followed simple cognitive principles to establish several rules that a reasonable difficulty control mechanism should satisfy:

**R1. Balance:** Correctly answering a more difficult question should gain a higher score. If the number of correctly answered *Easy* and *Hard* questions is equal, the level should be considered equivalent to the same total number of *Medium* questions. Additionally, the conventional range of the average score should correspond equally to the three difficulty levels in ascending order.

**R2. Generalizability:** The same formula, score gains, and thresholds should apply even to questions without difficulty labels, where difficulty is estimated solely based on the accuracy.

**R3. Improvability:** Correctly answering questions should increase the average score, and there should be a possibility of exceeding the threshold to advance to the next difficulty level.

**R4. Stability:** If the capability estimation remains at a certain difficulty for a long period, the likelihood of advancing to the next level should gradually decrease.

Following the above rules, we can analyze the mathematical conditions that $c_{it}$ and $t_{ij}$ must satisfy. From **R1**, we can get that $c_{1t} + c_{3t} = 2c_{2t} \wedge c_{1t} < c_{2t} < c_{3t}$. Since we have set the score for correctly answering a question without a difficulty level as $c_{2t}$, the range of $avg_s$ can be determined as $[0, c_{2t}]$, representing the spectrum from answering all questions incorrectly to answering all correctly. Based on the condition about threshold in **R1**, we can get that $t_{12} = c_{2t}/3$, $t_{23} = 2c_{2t}/3$ in the case that difficulty levels are absent. Due to **R2**, these threshold conditions can be generalized to the case where questions are provided with difficulty levels.

For **R3**, let $avg_s$ denote the average score after answering $N$ questions, and $avg_s'$ denote the average score after answering $N+1$ questions. Then **R3** is equivalent to satisfying the following conditions:

$$\textbf{C1.} \quad avg_s' = \frac{avg_s N + c_{it}}{N+1} > avg_s, \quad \text{holds true always when } avg_s \leq t_{(i,i+1)}$$

$$\textbf{C2.} \quad \exists\, avg_{s0}, N_0, \quad \text{s.t. } avg_s' = \frac{avg_{s0} N_0 + c_{it}}{N_0 + 1} > t_{(i,i+1)} \tag{2}$$

Simplifying Eq 2.C1, we find that $c_{it} > avg_s$ holds true always when $avg_s \leq t_{(i,i+1)}$. Therefore, we can conclude that $c_{it} > t_{(i,i+1)}$, meaning $c_{2t} > c_{1t} > t_{12}$ and $t_{23} < c_{2t} < c_{3t}$. Simplifying Eq 2.C2, we can obtain a relationship between $avg_{s0}$ and $N_0$:

$$avg_{s0} > t_{(i,i+1)} - \frac{c_{it} - t_{(i,i+1)}}{N_0} \tag{3}$$

Combining the condition $c_{it} > t_{(i,i+1)}$ from Eq 2.C1, this relationship indicates that, under the premise of **R3**, advancing to the next difficulty level by answering questions correctly requires a higher $avg_s$ as $N$ increases. This description is essentially **R4**: the longer one remains at a specific difficulty level, the harder it becomes to advance to the next level. Therefore, **R3**$\Rightarrow$**R4** is true.

Finally, **R4** can be stated as follows: given the average score $a \leq t_{(i,i+1)}$ after $N$ questions, the number $n$ of consecutive questions that must be answered correctly to surpass the level threshold should increase with $N$. We can obtain the following inequality:

$$\frac{aN + c_{it}n}{N+n} > t_{(i,i+1)} \Rightarrow (c_{it} - t_{(i,i+1)})n > (t_{i,i+1} - a)N \tag{4}$$

If **R4** holds, which means $n$ should increase as $N$ increases, combined with the inherent condition $a \leq t_{(i,i+1)}$, it follows that $c_{it} > t_{(i,i+1)}$, from which R3 can be deduced. Therefore, **R3**$\Rightarrow$**R4** and **R4**$\Rightarrow$**R3** hold simultaneously, meaning **R4** and **R3** are equivalent, demonstrating that we have now analyzed all the mathematical conditions that $c_{it}$ and $t_{(i,i+1)}$ must satisfy:

$$\begin{aligned}
&c_{1t} < c_{2t} < c_{3t} \\
&c_{1t} + c_{3t} = 2c_{2t} \\
&t_{12} = c_{2t}/3, \ t_{23} = 2c_{2t}/3 \\
&c_{it} > t_{(i,i+1)} \Rightarrow c_{2t} > c_{1t} > t_{12}, \ t_{23} < c_{2t} < c_{3t}
\end{aligned} \tag{5}$$

Therefor, we first set $c_{2t} = 1.5$, and so $t_{12} = 0.5$, $t_{23} = 1$. Then, since $1.5 = c_{2t} > c_{1t} > t_{12} = 0.5$ and $1 = t_{23} < c_{2t} = 1.5 < c_{3t}$, we set $c_{1t} = 1$ for convenience, and so $c_{3t} = 2c_{2t} - c_{1t} = 2$.

## C  CONSTRUCTION OF CONTEXT GRAPH

For the dataset selected during the Benchmark Grading stage, the process of constructing a context graph from its knowledge base (e.g., the set of all reference texts) can be divided into three components: text chunking, node construction, and node linking.

**Text Chunking:** To preserve the integrity of knowledge, the granularity of chunking is not refined to the sentence level. Instead, several sentences or an entire paragraph are grouped to form a single unit, treated as the minimal hierarchical "chunk". On this basis, chunks are further aggregated into higher-level "documents" according to whether they belong to the same article or serve as reference texts for the same question.

**Node Construction:** An LLM is employed to extract entities from each chunk, and the detailed prompt is provided in Appendix G. Each extracted entity serves as the subject for constructing a node. All chunks containing the same entity are assigned to the corresponding entity node. Formally, given an entity $e$, along with the chunks containing the entity $\{c_1, c_2, ...\}$ and the documents containing the entity $\{d_1, d_2, ...\}$, the node in the context graph can be defined as follows:

$$N = (e\,,\,\mathcal{C}\,,\,\mathcal{D})\,,\text{ where } \mathcal{C} = \{c_1, c_2...\}\,,\,\mathcal{D} = \{d_1, d_2, ...\} \tag{6}$$

In practice, we store only the IDs of chunks and documents within each node to conserve space.

**Node Linking:** During construction, if two entities appear together in the same chunk or the same document-level texts, their corresponding nodes in the context graph are treated as neighbors. Formally, given two nodes $N_1 = (e_1\,,\,\mathcal{C}_1\,,\,\mathcal{D}_1)$ and $N_2 = (e_2\,,\,\mathcal{C}_2\,,\,\mathcal{D}_2)$, if $\exists\ c_{1i} \in \mathcal{C}_1, c_{2j} \in \mathcal{C}_2$, s.t. $c_{1i} \equiv c_{2j}$ or $\exists\ d_{1i} \in \mathcal{D}_1, d_{2j} \in \mathcal{D}_\in$, s.t. $d_{1i} \equiv d_{2j}$, then $N_1$ and $N_2$ will be treated as neighbors in the context graph.

## D  STATISTICS AND PREPROCESSING OF DATASETS

### D.1  DETAILS OF DATASETS

The following benchmarks are used in our experiments, whose details are shown in Table 4.

**MedQA**(Jin et al., 2021) contains multiple-choice questions in the style of the Medical Licensing Examination. Questions in this dataset are collected from medical board exams in the US, Mainland China, and Taiwan, where human doctors are evaluated on their professional knowledge and ability to make clinical decisions. The background knowledge texts of MedQA are provided in the form of additional complete articles, and the questions only provide meta information.

**MultiHop-RAG**(Tang & Yang, 2024) consists of phrase Q&A queries, their ground truth answers, and the associated supported evidence constructed from news articles published between September and December 2023. The background knowledge texts of MultiHop-RAG are provided as supporting evidence along with the questions.

**QuALITY**(Pang et al., 2022) is a multiple-choice question dataset for long document comprehension, whose questions are written and validated by human contributors based on the long passages. The sources of QuALITY include: (1) Project Gutenberg fiction stories, which are mostly science fiction; (2) Slate magazine articles from the Open American National Corpus; (3) other nonfiction articles taken from The Long+Shor, Freesouls, and the book Open Access. QuALITY is organized by articles, with each data item consisting of a long article and several related questions, and the article serves as the background knowledge for each question.

### D.2  PREPROCESSING OF DATASETS

To verify Agent-as-Interviewer's ability to evaluate knowledge deficiencies in target models, we remove the background knowledge of the questions from MedQA and MultiHopRAG during evaluation. Otherwise, it would only test the target's reading comprehension ability rather than knowledge deficiencies. In contrast, we provided the background text when using QuALITY for evaluation, as the questions in QuALITY are highly dependent on the text content, and many of the texts are fictional narratives. Therefore, we use QuALITY to validate the JudgeAgent's effectiveness in guiding comprehension and reasoning rather than discovering knowledge deficiencies.

Table 4: Details of datasets in our experiments. The language of all the datasets is English.

| Datasets | Question Type | Categories | Splits | Used Splits | Split Size |
|----------|---------------|------------|--------|-------------|------------|
| MedQA | multiple-choice | medical
clinical | train
validation
test | test | 1273 |
| MultiHopRAG | phrase QA | technology
entertainment
sports
science
business
health | test | test | 2556 |
| QuALITY | multiple-choice | fiction stories
magazine articles
long articles | train
validation
test | validation | 2086 |

Additionally, we have reprocessed the difficulty levels of the questions from QuALITY. Based on the accuracy of human annotators in answering questions, QuALITY originally classified questions into two levels, *Easy* and *hard*, using a 50% accuracy threshold. To align with the difficulty levels defined by JudgeAgent's difficulty control module (*Easy*, *Medium*, and *Hard*), we re-labeled the questions using the same accuracy criteria. Questions of QuALITY with an accuracy below 1/3 are re-labeled as *Easy*, those with an accuracy between 1/3 and 2/3 as *Medium*, and the rest as *Hard*.

# E  ADDITIONAL EXPERIMENT ANALYSIS

## E.1  THE RESILIENCE AGAINST DATA CONTAMINATION

**Can Agent-as-Interviewer mitigate the challenges of data contamination in static benchmarking paradigms?**  In this section, we simulate a scenario where the static benchmarking evaluation paradigm suffers from data contamination by deliberately exposing the evaluation questions to LLMs during their training process. We selected Llama3-8B-Instruct, Mistral-7B-Instruct-v0.3, and Qwen2.5-7B-Instruct as base models, and constructed supervised fine-tuning (SFT) data from the MedQA and MultiHop-RAG benchmarks, which are intended for evaluation in the main experiments. These training data were used to fine-tune the selected base models. By comparing the performance differences between the original base models and the fine-tuned models on both the static benchmark questions and the extended questions generated by JudgeAgent, we analyze and verify the resilience of the Agent-as-Interviewer paradigm and its derivative JudgeAgent against data contamination.

Specifically, the procedure can be summarized as follows: for a base LLM $\mathcal{M}$ and a static evaluation benchmark $\mathcal{D}$, a fine-tuned LLM $\mathcal{M}$-sft is obtained by supervised fine-tuning with the training data constructed from $\mathcal{D}$. The performance of $\mathcal{M}$ and $\mathcal{M}$-sft, which is measured by the accuracy (ACC) in answering the questions, is then compared on both $\mathcal{D}$ and the extended questions $\mathcal{D}@K$ generated at the K-th iteration. The severity of data contamination ($\Delta$) is measured by the improvement in performance from the fine-tuned model $\mathcal{M}$-sft to the base model $\mathcal{M}$, which is formalized as $\Delta = ACC_{\mathcal{M}\text{-sft}} - ACC_{\mathcal{M}}$. To mitigate the effects of the LLM's randomness, each question was answered by the LLM 5 times. If the LLM produced a correct answer in three or more of these trials, it was considered to have answered the question correctly. We conduct our experiments on an Ubuntu machine with one 40GB NVIDIA A100 GPU. The results are shown in Table 5.

The experiment results indicate that, across various LLMs and benchmarks, fine-tuning with evaluation data leads to a notable improvement in model performance on base questions ($\Delta$-base), particularly evident on the MultiHop-RAG benchmark. These findings underscore the risks of data contamination: even when the original model exhibits limited performance on benchmarks, exposure to the benchmark evaluation data can artificially inflate its performance. Consequently, the

Table 5: The results for validating the resilience against data contamination. *base* means the performance on base static benchmark questions, and @*K* means extended questions at the K-th interation.

| Models | MedQA | | | | MultiHop-RAG | | | |
|---|---|---|---|---|---|---|---|---|
| | base | @1 | @2 | @3 | base | @1 | @2 | @3 |
| Llama3-8B-Instruct | 62.17 | 69.94 | 67.72 | 69.77 | 48.77 | 28.30 | 28.38 | 29.15 |
| Llama3-sft | 82.53 | 65.59 | 65.80 | 65.42 | 88.45 | 29.59 | 29.19 | 29.79 |
| Δ | **20.34** | -4.35 | -1.92 | -4.35 | **39.69** | 1.29 | 0.81 | 0.65 |
| Mistral-7B-Instruct-v0.3 | 57.91 | 59.40 | 58.76 | 56.97 | 59.51 | 37.75 | 38.60 | 37.22 |
| Mistral-sft | 62.73 | 53.18 | 51.98 | 51.98 | 92.17 | 32.74 | 32.58 | 32.62 |
| Δ | **4.82** | -6.23 | -6.78 | -4.99 | **32.66** | -5.01 | -6.02 | -4.60 |
| Qwen2.5-7B-Instruct | 85.93 | 78.04 | 75.44 | 77.31 | 46.31 | 20.59 | 21.64 | 21.48 |
| Qwen2.5-sft | 94.20 | 78.00 | 75.78 | 77.27 | 88.90 | 26.60 | 25.56 | 24.75 |
| Δ | **8.27** | -0.04 | 0.34 | -0.04 | **42.59** | 6.02 | 3.92 | 3.27 |

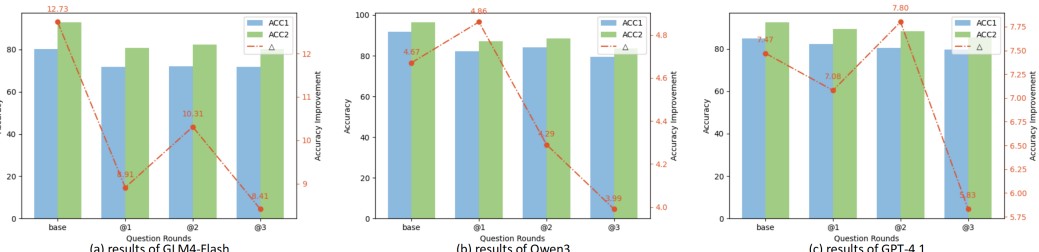

Figure 5: The cross-validation results of extended questions versus seed questions on MedQA. ACC1 and ACC2 indicate the accuracy before and after evaluation. Δ = ACC2 − ACC1 refers to the overall accuracy improvement. All the values are percentages. @K represents the results of the questions expanded at the K-th round.

model's genuine capabilities may be obscured by overestimated benchmark performance, leading to misapplication in scenarios beyond the actual capabilities.

In contrast, when evaluated on extended questions generated by JudgeAgent, the fine-tuned models and base models show little difference in performance. Additionally, fine-tuning even resulted in a decline on the MedQA benchmark. These results suggest that under the Agent-as-Interviewer dynamic evaluation paradigm, the generated questions maintain the validity of evaluation, even when the original questions have been exposed to the target model. Furthermore, under the same setting of LLM and benchmark, there is a marked gap between the performance gain on base questions and extended questions after fine-tuning, demonstrating that Agent-as-Interviewer exhibits considerable resilience to data contamination.

### E.2 CROSS VALIDATION OF THE EVALUATION SUGGESTIONS

**Do the suggestions provided by Agent-as-Interviewer only take effect for the seed questions?** To address this question, we designed cross-validation experiments from two perspectives.

First, considering that the evaluation suggestions are derived from a comprehensive evaluation of the responses to both the seed questions and their extended questions, we evaluated and compared the accuracy improvement of the suggestions on the extended questions versus the seed questions, aiming at verifying that the suggestions do not contain "cheating information" specific to seed questions in the scenario after evaluation. The results are shown in Figure 5.

Secondly, we transferred the evaluation suggestions derived from seed questions to other questions with related knowledge concepts in the benchmark, to verify the effectiveness of the suggestions within the same knowledge domain. Specifically, we categorized questions based on the knowledge entities they contain, formalized as follows: given the context graph $\mathcal{G}$, for two questions $q_1$ and $q_2$ with knowledge entities $E_1 = \{e|e \in q_1 \land e \in \mathcal{G}\}$ and $E_1 = \{e|e \in q_2 \land e \in \mathcal{G}\}$, if $E_1$ and $E_2$ have

a non-empty intersection, then $q_1$ and $q_2$ are considered related questions. In this experiment, the suggestions for a question were constructed from the suggestions of its related questions, excluding suggestions from the question itself, to assess and validate the transferability of the suggestions provided by the Agent-as-Interviewer. We screened out questions without relevant questions and those with only relevant questions based on non-knowledge entities, such as male, female, 2 years, etc. The results are shown in Table 6.

Table 6: The cross-validation results of transferring suggestions to related questions. All the values are percentages. *Non-transfer* refers to suggestions being applied to the seed questions, whereas *Transfer* refers to them being applied to related questions.

| Target | ACC1 | Non-transfer | | | | Transfer | | | |
|---|---|---|---|---|---|---|---|---|---|
| | | ACC2 | CR↑ | CtE↓ | Δ↑ | ACC2 | CR↑ | CtE↓ | Δ↑ |
| GLM4-Flash | 78.42 | 87.49 | 18.30 | 9.23 | 9.07 | 85.56 | 16.98 | 9.84 | 7.13 |
| GPT-4.1 | 84.21 | 91.09 | 10.23 | 3.35 | 6.88 | 89.78 | 9.50 | 3.93 | 5.57 |
| Qwen3 | 92.44 | 96.06 | 5.96 | 2.34 | 3.62 | 95.17 | 5.17 | 2.44 | 2.73 |

First, we analyze the difference in the effectiveness of evaluation suggestions on seed questions versus extended questions. As shown in Figure 5, suggestions effectively improve performance for both seed questions and extended questions, with a relatively small gap in the degree of improvement. Notably, when Qwen3 and GPT-4.1 are used as target models, the first round of Qwen3 and the second round of GPT-4.1 exhibit even greater improvement than seed questions. These results indicate that although the suggestions only supplement knowledge for several key concepts, such as the case in Figure 7, such concise suggestions can still benefit both seed and extended questions, demonstrating that Agent-as-Interviewer is capable of identifying and addressing knowledge gaps in the target model, rather than simply providing "cheating information" specific to seed questions.

Furthermore, it is observed that the effectiveness of suggestions for third-round questions is consistently low in the experiments. This may be because, by the third round, knowledge path sampling has expanded beyond the scope of knowledge related to seed questions to a broader range. As a result, the generated questions diverge more significantly in core knowledge from earlier questions, thereby reducing the effectiveness of the knowledge guidance provided in the suggestions.

Next, we analyze the difference in the effectiveness of suggestions on questions that share the same knowledge concepts. As shown in Table 6, compared to their effectiveness on seed questions, the suggestions exhibit a slight decrease when applied to related questions, as indicated by a decline in CR and Δ, and an increase in CtE. But the difference is minor, and the improvement remains notable, suggesting that the suggestions can be effectively transferred to other questions involving the same knowledge concepts. This further demonstrates that Agent-as-Interviewer can provide suggestions that do not simply serve as "cheating information" specific to seed questions.

However, the slight decline in evaluation effectiveness also indicates that the knowledge guidance provided in the suggestions is not fully aligned with the related questions. This may be because the overlapping entities between these related questions and seed questions do not correspond to core knowledge concepts. For example, suggestions centered on "blood type" may be transferred to a question where "serum" is the core knowledge concept, resulting in a partial mismatch.

The above experiment results demonstrate that the evaluation suggestions provided by Agent-as-Interviewer are not only applicable to seed questions but can also be transferred to other questions that share relevant core knowledge concepts.

### E.3 SUPPLEMENTARY PARAMETER ANALYSIS

**What is the impact of batch size in the *Benchmark Grading* stage on the evaluations of the Agent-as-Interviewer?** In *Benchmark Grading* stage, questions are divided into batches to comprehensively assess the target's capabilities in a base level. These batches are also the basic units for question extension and evaluation feedback. Given a fixed rounds, the batch size is inversely proportional to the number of batches, extended questions, and evaluation suggestions, thereby influencing the time and resource consumption of the entire evaluation process. **Can the batch size be**

**maximized to reduce resource consumption while maintaining the effectiveness of evaluations?**
To address this question, we conducted a parameter analysis experiment, examining the evaluation
effectiveness and time consumption under different batch sizes. The results are shown in Figure 6.

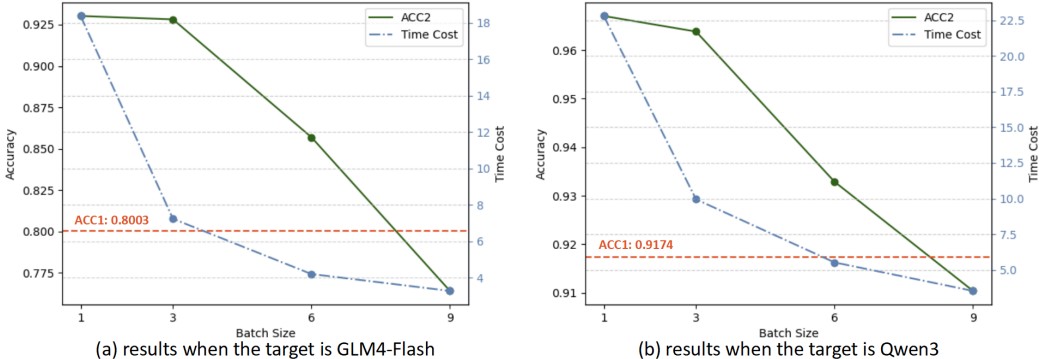

(a) results when the target is GLM4-Flash          (b) results when the target is Qwen3

Figure 6: The results of different batch sizes on MedQA. ACC1 and ACC2 indicate the accuracy
before and after receiving evaluations. Time Cost is the average time consumption for each question.

As observed from the trend of the curves in Figure 6, both the target's accuracy after receiving evalu-
ation suggestions (ACC2) and the average time consumption per question for evaluation (Time Cost)
decrease as the batch size increases. Among them, the decline rate in ACC2 gradually accelerates
with larger batch sizes, while the decline rate in Time Cost gradually slows, exhibiting a marginal
effect. The reason for the marginal effect in Time Cost lies in the fact that the time required for
the target model to answer the seed questions, which is a component of the overall evaluation pro-
cess, varies little with changes in batch size. As a result, there is a threshold beyond which further
reductions in time cost have diminishing returns.

The decline in ACC2 with increasing batch size can be attributed to the expansion of the question's
knowledge domain. As the batch size grows, the generated questions during evaluation become
more heterogeneous and less coherent with the knowledge relevant to seed questions, making it
difficult for JudgeAgent to identify appropriate knowledge guidance from the dispersed question-
answer pairs. Moreover, when the batch size exceeds the number of extension rounds, the disorder
of knowledge scopes intensifies more rapidly, ultimately leading to a sharp drop in accuracy. The
results in Figure 6 show that when the batch size reaches 9, the evaluation suggestions even become
counterproductive (ACC2 < ACC1), interfering with the normal reasoning of the target model.

Therefore, considering both the evaluation time cost and effectiveness, we selected a batch size of 3
in our experiments as a balanced choice.

## F  ALGORITHM

To clarify the entire workflow of JudgeAgent, we use pseudocode to show the dynamic evaluation
process in Algorithm 1, the validation process of evaluation in Algorithm 2, the construction of
context graph in Algorithm 3, and the generation of extended questions in Algorithm 4.

## G  PROMPTS

We show the detailed prompt for Entity Extraction in Prompt 7, the prompt for generating extended
questions in Prompt 8, the prompt for evaluating the performance of the target LLM in Prompt 9,
and the prompt for querying the target LLM with suggestions in Prompt 10.

## H  DETAILED CONTENT OF CASE STUDY

The detailed content of the comparative case in section 4.6 is shown in Figure 7.

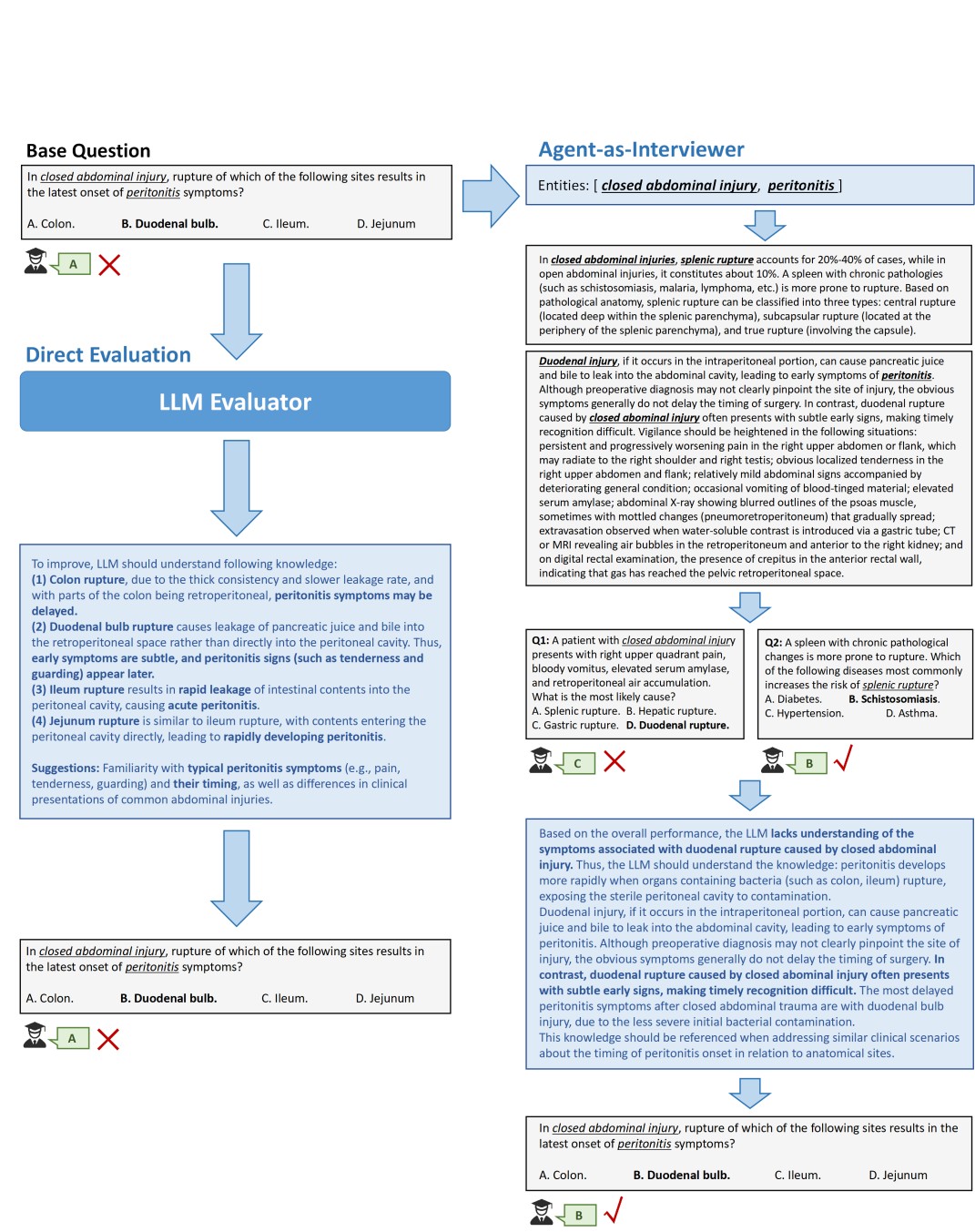

Figure 7: The detailed content of the case.

---

**Algorithm 1** Dynamic evaluation process of JudgeAgent

---

1: **Input:** Target LLM $\mathcal{M}_t$, Base dataset $\mathcal{D} = \{(q_i, a_i, d_i)\}_{i=1}^N$ (each item include question $q$, answer $a$, and difficulty $d$), Knowledge bases $\mathcal{K} = \{k_1, k_2, ...\}$, Core LLM of JudgeAgent $\mathcal{M}_c$, Predefined batch capacity $N_B$, Max extension round $RND_e$, Max hop of sampling $H$

2: **Output:** Evaluation Score $sc$, Batched dataset with suggestions $\mathcal{D}_S = \{(\mathcal{B}_1, s_1), \dots\}$
   *// Construct context graph*

3: $\mathcal{G} \leftarrow$ CONSTRUCT_CONTEXT_GRAPH($\mathcal{K}, \mathcal{M}_c$)
   *// Split base dataset into batches*

4: $\{\mathcal{B}_1 = \{(q_{1i}, a_{1i}, d_{1i})\}_{i=1}^{N_B}, \dots\} \leftarrow$ SPLIT_BATCHES($\mathcal{D}, N_B$)
   *// Begin Evaluation*

5: $sc \leftarrow 0$, $N_{total} \leftarrow 0$

6: $\mathcal{D}_S \leftarrow \{\}$

7: **for** $\mathcal{B} \leftarrow \{\mathcal{B}_1, \mathcal{B}_2, \dots\}$ **do**

8:    $sc_\mathcal{B} \leftarrow 0$

9:    $RND_{total} \leftarrow 0$

10:    $\mathcal{Q}_{tested} \leftarrow \{\}$
    *// Stage1: Benchmark Grading*

11:    **for** $(q, a, d) \leftarrow \mathcal{B}$ **do**

12:       Get answer $a_\mathcal{M} \leftarrow$ QUERY_LLM($q, \mathcal{M}_t$)

13:       **if** $a_\mathcal{M}$ is correct based on $q$ and $a$ **then**

14:          $sc_\mathcal{B} \leftarrow sc_\mathcal{B}+$ DIFFICULTY_SCORE($d$)

15:       **end if**

16:       $RND_{total} \leftarrow RND_{total} + 1$

17:       Add $(q, a, d, a_\mathcal{M})$ to $\mathcal{Q}_{tested}$

18:    **end for**

19:    Decide difficulty $d_e \leftarrow$ DECIDE_DIFFICULTY($sc_\mathcal{B}, RND_{total}$)
    *// Stage2: Interactive Extension*

20:    **for** $i \leftarrow \{1, 2, \dots, RND_e\}$ **do**

21:       $(q_e, a_e, t) \leftarrow$ GENERATE_EXTENDED_QUESTIONS($\mathcal{B}, \mathcal{G}, \mathcal{M}_c, H, d_e$)

22:       Get answer $a_\mathcal{M} \leftarrow$ QUERY_LLM($q_e, \mathcal{M}_t$)

23:       **if** $a_\mathcal{M}$ is correct based on $q_e$ and $a_e$ **then**

24:          $sc_\mathcal{B} \leftarrow sc_\mathcal{B}+$ DIFFICULTY_SCORE($d_e$)

25:       **end if**

26:       $RND_{total} \leftarrow RND_{total} + 1$

27:       Add $(q_e, a_e, d_e, a_\mathcal{M})$ to $\mathcal{Q}_{tested}$

28:       Decide difficulty $d_e \leftarrow$ DECIDE_DIFFICULTY($sc_\mathcal{B}, RND_{total}$)

29:    **end for**
    *// Stage3: Evaluation Feedback*

30:    Get evaluation suggestions $s \leftarrow$ EVALUATE($\mathcal{Q}_{tested}, \mathcal{M}_c$)

31:    Add $(\mathcal{B}, s)$ to $\mathcal{D}_S$

32:    $sc \leftarrow sc + sc_\mathcal{B}$

33:    $N_{total} \leftarrow N_{total} + RND_{total}$

34: **end for**

35: $sc \leftarrow sc / N_{total}$

36: **return** $sc, \mathcal{D}_S$

---

**Algorithm 2** Validation process of evaluation results from JudgeAgent

1: **Input:** Target LLM $\mathcal{M}_t$, Batched dataset with suggestions $\mathcal{D}_S = \{(\mathcal{B}_1, s_1), \dots\}$, in which $\mathcal{B}_i = \{(q_{i1}, a_{i1}, d_{i1}), \dots\}$ is a batch of base dataset $\mathcal{D}$, and $\mathcal{S}_i = \{s_{i1}, \dots\}$ is relevant suggestions from JudgeAgent.

2: **Output:** Accuracy of target LLM before evaluation $acc_1$, Accuracy after evaluations $acc_2$, Correction Rate $cr$, Correct-to-Error Rate $ce$

    *// Initialize counter of questions*

3: $N_{acc1} \leftarrow 0$, $N_{acc2} \leftarrow 0$, $N_{cr} \leftarrow 0$, $N_{ce} \leftarrow 0$, $N_{total} \leftarrow 0$

4: **for** $(\mathcal{B}, s) \leftarrow \mathcal{D}_S$ **do**

5:     $N_\mathcal{B} \leftarrow$ LEN$(\mathcal{B})$

6:     $N_{total} \leftarrow N_{total} + N_\mathcal{B}$

7:     **for** $i \leftarrow \{1, 2, \dots, N_\mathcal{B}\}$ **do**

8:         $(q, a, d) \leftarrow \mathcal{B}[i]$

9:         Get answer1 $a_1 \leftarrow$ QUERY_LLM$(q, \mathcal{M}_t)$

10:       Get answer2 $a_2 \leftarrow$ QUERY_LLM_WITH_SUGGESTIONS$(q, s, \mathcal{M}_t)$

11:       correct1 $\leftarrow$ whether $a_1$ is correct based on $q$ and $a$

12:       correct2 $\leftarrow$ whether $a_2$ is correct based on $q$ and $a$

13:       **if** correct1 **then**

14:           $N_{acc1} \leftarrow N_{acc1} + 1$

15:           **if** not correct2 **then**

16:             $N_{ce} \leftarrow N_{ce} + 1$

17:           **end if**

18:       **end if**

19:       **if** correct2 **then** State $N_{acc2} \leftarrow N_{acc2} + 1$

20:           **if** not correct1 **then**

21:             $N_{cr} \leftarrow N_{cr} + 1$

22:           **end if**

23:       **end if**

24:     **end for**

25: **end for**

26: $acc_1 \leftarrow N_{acc1}/N_{total}$, $acc_2 \leftarrow N_{acc2}/N_{total}$, $cr \leftarrow N_{cr}/N_{total}$, $ce \leftarrow N_{ce}/N_{total}$

27: **return** $acc_1$, $acc_2$, $cr$, $ce$

---

**Prompt for Entity Extraction**

Please identify and label the entities in the following multiple sentences, and return the entity labeling results for each sentence.

The results for each sentence should be independent, in JSON format, containing the sentence number, sentence text, and the list of recognized entities (including entity text, type, and position).

Return format is a dictionary, with only one key 'labeled_data', and the value is a list, each element is a dictionary containing the sentence text and the entity list.

```
{{
    "labeled_data":
    [
        {{"text":"Sentence 1", "entity_list": [{{"entity_text": "", "entity_type": ""}}]}},
        {{"text":"Sentence 2", "entity_list": [{{"entity_text": "", "entity_type": ""}}]}},
        ...
    ]
}}
```

Notice that "text" should be only the sentence, not the whole article. Sentence list:

{ *Sentences* }

Table 7: Prompt for entity extraction.

---

**Algorithm 3** Construction process of context graph

---

1: **Input:** Knowledge Base of Dataset $\mathcal{K} = \{k_1, k_2, \dots\}$, an LLM $\mathcal{M}$
2: **Output:** Context Graph $\mathcal{G} = (\mathcal{N}, \mathcal{E})$, in which $\mathcal{N}$ is the node set, and $\mathcal{E}$ is the edge set.
3: $\mathcal{N} \leftarrow \{\}$, $\mathcal{E} \leftarrow \{\}$     *// Initialize node set and edge set as empty dictionary*
4: **for** $k \leftarrow \mathcal{K}$ **do**
    *// Spliting text into chunks*
5:     $C = \{c_1, c_2, \dots\} \leftarrow$ SPLIT_TO_CHUNKS$(k)$
    *// Prompting LLM to label entities*
6:     $P_e \leftarrow \emptyset$
7:     **for** $c \leftarrow \{c_1, c_2, \dots\}$ **do**
8:       $S_e = \{e_1, e_2, \dots\} \leftarrow$ LABEL_ENTITY$(c, \mathcal{M})$
9:       **for** $e \leftarrow S_e$ **do**
10:        **if** $e$ not in $\mathcal{N}$ **then**
11:         $\mathcal{N}[e] \leftarrow (e, \mathcal{C} = \emptyset, \mathcal{D} = \emptyset)$
12:         $\mathcal{E}[e] \leftarrow \emptyset$
13:        **end if**
14:        Add $c$ to set $\mathcal{N}[e].\mathcal{C}$
15:        Add $k$ to set $\mathcal{N}[e].\mathcal{D}$
16:        Expand set $\mathcal{E}[e]$ with $S_e$
17:       **end for**
18:       Expand set $P_e$ with $S_e$
19:     **end for**
20:     **for** $e \leftarrow P_e$ **do**
21:       Expand set $\mathcal{E}[e]$ with $P_e$
22:     **end for**
23: **end for**
24: $\mathcal{G} \leftarrow (\mathcal{N}, \mathcal{E})$
25: **return** $\mathcal{G}$

---

**Algorithm 4** Generation process of extended questions

---

1: **Input:** Base Question $\mathcal{Q} = \{q_1, q_2, \dots\}$, Context Graph $\mathcal{G} = (\mathcal{N}, \mathcal{E})$, LLM $\mathcal{M}$, Max hop of path $H$, Difficulty $d$
2: **Output:** Extended question with its answer and background text $(q_e, a_e, t)$
    *// Prompting LLM to label entities from $\mathcal{Q}$*
3: $S_e \leftarrow \{\}$
4: **for** $q \leftarrow \mathcal{Q}$ **do**
5:     $set_e = \{e_1, e_2, \dots\} \leftarrow$ LABEL_ENTITY$(q, \mathcal{M})$
6:     Add $set_e$ to $S_e$
7: **end for**
8: $e \leftarrow$ RANDOM_SAMPLE$(\{e_1, e_2, \dots\}, 1)$
    *// Sample knowledge paths*
9: $e' \leftarrow$ the most similar entity in $\mathcal{N}$ of $\mathcal{G}$
10: $E_{visited} \leftarrow \{e'\}$
11: $t \leftarrow$ RANDOM_SAMPLE$(\mathcal{N}[e'].\mathcal{C}, 1)$
12: **for** $i \leftarrow \{1, 2, \dots, H-1\}$ **do**
13:     $E_{candidate} \leftarrow$ the most similar 5 entities to $e'$ in $\mathcal{E}[e']$ that not in $E_{visited}$
14:     $e' \leftarrow$ RANDOM_SAMPLE$(E_{candidate}, 1)$
15:     $C_{candidate} \leftarrow$ the most similar 5 chunks to $q$ in $\mathcal{N}[e'].\mathcal{C}$
16:     $c \leftarrow$ RANDOM_SAMPLE$(C_{candidate}, 1)$
17:     Concatenate $c$ to new line of $t$
18:     Add $e'$ to $E_{visited}$
19: **end for**
20: $(q_e, a_e) \leftarrow$ GENERATE_QUESTION$(t, \mathcal{M})$
21: **return** $(q_e, a_e, t)$

---

**Prompt for Generating Extended Questions**

As an interviewer, you are tasked with designing questions based on the provided texts. Your role involves crafting questions and correct answers that fulfill the following criteria:

1. **Focus on the Entity**: Ensure all questions consistently center around the specified entity from the article.
2. **Ensure Accuracy and Conciseness of Answers**: Verify that the provided answer is both correct for your designed question within the context and logic of the given text fragments, and ensure the answer is sufficiently concise—presented as a word or phrase, avoiding redundancy.
3. **Conform to difficulty requirements**: You need to design questions for the required difficulty levels, with specific requirements as follows:

    (1). [easy]: **Encourage Knowledge Memorization**, design questions that assess whether respondents have memorized relevant knowledge. Create questions by directly extracting and blanking out content from the given passage.

    (2). [medium]: **Encourage Knowledge Comprehension**: Design questions that prompt respondents to dissect and comprehend concepts involved in the topic. Avoid assessing only superficial knowledge retention.

    (3). [hard]: **Encourage Knowledge Deep Analysis**: Design questions that prompt respondents to engage in deep thinking and analysis. Avoid merely testing knowledge recall or conceptual comprehension; do not simply extract fragments from the given passage to create fill-in-the-blank items. Encourage respondents to focus on entities within the question and employ logical skills for complex reasoning.

Here are examples:
{ *Examples* }

Now, given the following text fragments:
{ *context* }

Based on the provided texts, please generate questions by following the requirements above and referencing the examples.
Output in the specified JSON format below: {{
    "generated_question":
    [
        {{
            "question": "Generated Question",
            "answer": "Correct Answer of Generated Question"
        }},
        ...
    ]
}}

Table 8: Prompt for generating extended questions based on different difficulties.

**Prompt for Evaluation**

The following is the performance of an LLM in answering a series of questions:

{ *list of questions, correct answers, and LLM's answers* }

Please evaluate and analyze the interviewee's performance based on the above performance using concise language from the following perspectives, and provide suggestions that help the LLM answer the same questions better. Suggestions should provide specific and detailed guidance on logical thinking steps, required knowledge, and abilities, ensuring the LLM can answer correctly for the same questions.
Output in the following JSON format:
{{
    "flaws_knowledge": "The lack of background knowledge.",
    "flaws_capability": "The flaws in logic and capability.",
    "comprehensive_performance": "The Comprehensive performance of all questions.",
    "suggestions": "Suggestions that help the LLM answer questions bette"
}}

Table 9: Prompt for evaluating the performance of the target LLM.

**Prompt for Querying LLM with suggestions**

Please complete the following question:
[question]: { *question* }

In your previous responses to these questions, the interviewer has provided the following suggestions for you to help you answer better:
[suggestions]: { *suggestions* }

Please consider the above [suggestions], and answer the above [question], in the following JSON format:
{{"answer": "Your answer"}}

Table 10: Prompt for querying the target LLM with suggestions.

