# OpenReview forum: "JudgeAgent: Knowledge-wise and Dynamic LLM Evaluation with Agent-as-Interviewer"
_ICLR.cc/2026/Conference — ICLR 2026 Conference Withdrawn Submission_

### Official Review · Reviewer_Fk12 · 2025-10-27

**Soundness:** 1
**Presentation:** 2
**Contribution:** 2
**Rating:** 2
**Confidence:** 4

**Summary:**

This work proposes the JudgeAgent framework for dynamic evaluation of model knowledge, summarized as follows:
(1) Evaluate on static base datasets to determine model proficiency.
(2) Extend from seed questions using LLM entity extraction + graph tools to other similar entities and reference texts, then generate new questions via LLMs at difficulty levels matching the model's proficiency, with difficulty estimation subject to iterative refinement; the questions are dynamic, reducing data-leakage risk of static benchmarks.
(3) LLMs provide evaluation feedback based on their question-answering history; the evaluation paradigm further supplies valuable suggestions to the model, helping it self-optimize beyond knowledge association and reasoning.

Experiments demonstrate that this framework's feedback enhances model performance on static base datasets; the authors also quantify how question difficulty changes and supply theoretical analysis.

**Strengths:**

(1) The paper's expressions and figures are clear.
(2) The framework design is clear and easy to understand.
(3) The authors conducted experiments using multiple models from different departments, enhancing reliability.
(4) Proposes using dynamically difficult questions to test LLMs to capture model capabilities more accurately.
(5) Introduces a knowledge-based data expansion scheme in the evaluation system to reduce the risk of data leakage.

**Weaknesses:**

After reviewing the experimental section, the connection between the motivation for the work and the experimental design is perplexing. As an evaluation framework, JudgeAgent's key improvements over previous dynamic evaluation frameworks lie in its broader knowledge coverage and difficulty control; the primary motivation should be to mitigate evaluation bias within the framework and avoid lack of discriminative power due to difficulty mismatch (i.e., to propose a more precise evaluation paradigm). However, the experiments primarily focus on demonstrating that the feedback provided by the framework can improve model performance on base datasets.

(1) No comparison with other evaluation paradigms (e.g., scope limitations, manual accuracy); instead, the focus is on describing that this evaluation paradigm can provide feedback to improve the model's score on the benchmark.
(2) There are no concrete metrics quantifying model performance under this evaluation framework, nor analyses of evaluation bias or discriminative power relative to other frameworks.
(3) Whether feedback improves model performance on baseline questions has no direct correlation with evaluation accuracy, since the feedback is generated in an open-book manner and numerous factors can influence model performance; the improvement in benchmark score amounts to checking for omissions and optimizing the LLM, which does not represent any effective meaning.
(4) If the authors' motivation is "to enhance model performance in knowledge question-answering through feedback," the inconsistency lies in the fact that the entire process requires a complete benchmark and answers for grading, and any eventual improvement from feedback should occur offline by expanding the model's knowledge boundaries through training; however, the framework ultimately provides feedback in an online scenario by stating it within the prompt to the model. In actual online scenarios this framework cannot be used to enhance performance.
(5) The dynamic-difficulty aspect resembles merely an instruction evolution of the benchmark; context learning is performed based on new instructions and results, which loses the original purpose of dynamic difficulty—to capture the LLM's capabilities more accurately.

Overall, the authors need to clearly articulate the framework's motivation and make corresponding revisions to the writing and experimental design.

**Questions:**

(1) Are manual reviews conducted to ensure that LLMs do not generate cheating information directly related to the seed question during the feedback evaluation phase (L248-249)?
(2) Why are 1/3 and 1/2 used in the difficulty threshold setting? Why is the difficulty expressed as linear?
(3) Is the work more focused on a new evaluation paradigm to accurately evaluate LLM performance, or is it more focused on how to improve the performance of LLM on a benchmark?

---

> ### Author Response · Authors · 2025-11-21
> **Responses to Reviewer Fk12, About the Weakneses**
>
> We sincerely thank Reviewer Fk12 for the helpful and valuable comments on our work.
>
> Firstly, we need to clarify that, **the core of our work is to establish a dynamic evaluation paradigm capable of precisely assessing the knowledge and capabilities of LLMs.**
>
> The previous works struggle to ensure controllability of evaluation questions and knowledge coverage, while also facing limited practical value of evaluation feedback caused by difficulty dismatch. To address these limitations and better meet real-world application needs, **we delegate content expansion and knowledge extension to a context graph. Through a controlled sampling mechanism, we ensure the accuracy and relevance of knowledge** (as described in **Sec 3.2, Lines 208-213**). The LLM is only responsible for extracting knowledge entities from questions and generating questions based on reliable knowledge, significantly reducing the impact of hallucinations or biases. Simultaneously, **a difficulty adjustment mechanism ensures that evaluation questions remain within the normal response range of the target model, guaranteeing the effectiveness of the evaluation feedback**. These aspects represent the core value of our work.
>
> As for why we validated our framework by observing that evaluation feedback can improve the model's performance on benchmarks, this is because existing works—whether benchmarks or evaluation methods—lack universally accepted quantitative metrics for evaluation effectiveness and do not facilitate mutual comparison. Considering that **the ultimate goal of evaluation is to optimize the target model's performance**, as explained in **Sec 3.3, Lines 244-247**, we verify the effectiveness of our evaluation method by observing whether the evaluation feedback can guide the target model to improve its performance. **To ensure relatively objective and quantifiable results**, we chose to measure the target model's improvement on existing benchmarks. **This is only a validation approach and not the core of our work.**
>
> Here is our response and clarification for the weaknesses.
>
> **Weakness1&2:** For the lack of comparison with other evaluation paradigms, **in Sec 4.4 Lines 398-403**, we conduct experiments on the effectiveness of static benchmarking paradigm (w/o interactive extension). The results in **Table 3** show that the evaluation enabled by the knowledge-driven and difficulty-control **dynamic expansion of JudgeAgent is crucial to the final evaluation efffectiveness**, whereas the **static baseline paradigm proves insufficient**. As for other dynamic paradigms, existing dynamic evaluation studies lack established quantitative standards and comparative analyses among themselves.
>
> **Weakness3:** For the relationship between feedback and benchmark performance, we **did not generate feedback in an open-book manner**. As mentioned in **Sec 3.3, Lines 245-246**, **no correct answers or other cheating information were provided during feedback generation.** Experiments in **Appendix E.2 (Pages 16-17)** also verified that **the evaluation feedback contains no cheating information** and is equally effective for seed questions and other questions with similar knowledge.
>
> As for the reviewer's point the the improvement in benchmark score amounts to checking for omissions and optimizing the LLM, it should be noted that **the purpose of evaluation is to use the evaluation results as a reference to identify omissions and shortcomings in the target model, thereby enabling targeted optimization**. In practical applications, the process of model optimization and iteration follows this same workflow: constructing evaluation tasks based on requirements to assess the target model, and then iteratively refining the model according to the evaluation results. **Using benchmarks is solely for the purpose of quantifying results more conveniently and objectively, and the validation process itself holds effective meaning.**
>
> **Weakness4:** For the question that performance improvements should be achieved through offline training, since **the target models in our experiments are all closed-source**, we cannot optimize them through offline training. So we adopted a method of generating feedback suggestions based on evaluation results and explicitly incorporating them into the prompt to guide the target model's answering process.
>
> **Weakness5:** For the dynamic difficulty design, we need to clarify a misunderstanding. **The new evaluation questions are not simply generated through in-context learning or evolution based on the original benchmark**. Instead, we expand retrieval on the contextual knowledge graph based on the knowledge in the original instructions. The retrieved background knowledge is then used to generate new evaluation questions through dynamic difficulty control, enabling a systematic and comprehensive assessment of the target large model's knowledge mastery.

---

> ### Author Response · Authors · 2025-11-21
> **Responses to Reviewer Fk12, About the Questions**
>
> Thanks for your questions. Here are our responses to these questions:
>
> **1. About the risk of cheating information:**
>
> Due to the lack of domain-specific personnel, we did not perform validation through manual review. However, we ensured the evaluation feedback contains no cheating information from two aspects:
>
> First, as mentioned in **Sec 3.3, Lines 245-246**, during feedback generation, only the question, the target model's response, and whether the response was correct are provided as information. **The correct answers and background knowledge were deliberately withheld, ensuring the evaluator Agent has no access to cheating information at the source.**
>
> Second, for whether the evaluation feedback contains cheating information, we have experimentally verified it in **Appendix E.2 (Pages 16-17)**. We applied the evaluation feedback to guide the target model in answering other questions with related knowledge. The results in **Table 6** demonstrate that the effectiveness of the evaluation on related-knowledge questions is comparable to that on seed questions, indicating that **the evaluation feedback does not serve as “cheating information” specific to seed questions.**
>
> **2. About the design of difficulty control:**
>
> First, we employ a linear metric to represent difficulty, inspired by the common real-world practice of scoring student exams. Questions of varying difficulty are assigned different score values, with correct answers earning scores, and the total score is used to assess the student's proficiency. Based on whether the student is high or low performing, the decision is made to adjust the difficulty of the exam. The advantages of this linear scoring and segmented differentiation lie in its relative fairness and objectivity, as well as the simplicity of cumulative calculation, which helps establish a quantitative and objective ability estimation and difficulty segmentation for the target model in dynamic evaluation.
>
> Second, for the assignment of scores and difficulty thresholds, we have provided a detailed explanation and justification of our difficulty adjustment mechanism, score assignments, and threshold settings in **Sec 3.1, Lines 184-192** and **Appendix B (Pages 12-13)**. For the sake of balance, we evenly divided the range of accuracy to ensure the three difficulty segments are relatively uniform.
>
> **3. About the focus of our work:**
>
> This work **aims to establish a dynamic evaluation paradigm capable of precisely assessing the knowledge and capabilities of LLMs.** The use of benchmark performance improvement as a metric in experiments is solely to provide a quantifiable and relatively objective validation method.
>
> Current works lacks a universally accepted method for validating the effectiveness of evaluation methods. Therefore, as described in **Sec 3.3, Lines 244–247**, we verify the effectiveness of our framework by observing whether the evaluation feedback can guide the target model to improve its performance. **This is only a validation approach and not the core of our work.** Our work **focus remains on refining existing dynamic evaluation paradigms through perspectives such as knowledge-driven strategies and difficulty adaptation.** How to optimize target models based on the evaluation results generated by our framework is planned for our future work.

---

> ### Author Response · Authors · 2025-11-25
>
> We sincerely appreciate the reviewers' valuable feedback. We have carefully addressed the concerns raised and hope that our responses have provided sufficient clarification. If you have any remaining questions that require further discussion, we are happy to discuss.
>
> Could you please let us know whether our rebuttal resolves your concerns? We look forward to your further comments and feedback.

---

> ### Comment · Reviewer_Fk12 · 2025-11-26
>
> Thanks for the authors' response. On lines 73-74, the authors propose two motivations: biased evaluation and mismatched difficulty. For an evaluator aiming to achieve precise evaluation, I believe these motivations (if their existence can be confirmed) make sense:
>
> - Biased evaluation. Uncontrolled use of LLMs for dynamic evaluation may introduce hallucinations or biases, leading to skewed evaluation results.
>
> - Mismatched difficulty. Using tasks with mismatched difficulty levels (e.g., overly simple questions for strong models or excessively difficult questions for weak models) may fail to yield meaningful information for evaluation.
>
> However, improvements on either front require direct evidence for validation. The claim that generated feedback enhances performance on original benchmarks constitutes overly indirect evidence, as the factors driving this performance boost remain uncertain. We cannot definitively link this improvement to evaluator precision. I understand that "lack universally accepted quantitative metrics for evaluation effectiveness" may pose a challenge. However, if the core objective is to propose an evaluator, we must establish clear and comparable metrics for evaluating the effectiveness of different evaluators. Otherwise, we cannot confirm whether the problem outlined in the motivation truly exists, nor can we fully demonstrate the effectiveness of the proposed evaluator.
>
>
> If JudgeAgent were proposed not as an evaluator but as a feedback generator for improvement, I would consider the current experimental design acceptable. However, Weakness 4 remains an issue in this case.
>
> I greatly appreciate the authors' efforts in their response, but since my primary concerns remain unaddressed, I maintain my current rating.

---

### Official Review · Reviewer_7GU8 · 2025-10-31

**Soundness:** 4
**Presentation:** 4
**Contribution:** 4
**Rating:** 6
**Confidence:** 4

**Summary:**

This paper proposes Agent-as-Interviewer, a novel dynamic evaluation paradigm for LLMs. Unlike static benchmark based evaluations, which often suffer from overestimation, limited scope, and fixed difficulty, this paradigm enables multi-turn, adaptive evaluations using LLM agents that dynamically generate and adjust questions based on the target model’s responses. The authors introduce JudgeAgent, a framework that consists of three core stages, 1. benchmark grading, 2. interactive extension, 3. evaluation feedback. Extensive experiments show that JudgeAgent can effectively identify capability gaps in LLMs and improve their performance through adaptive, knowledge-based feedback.

**Strengths:**

1. Introduces a multi-turn, dynamic evaluation paradigm enabling more adaptive and realistic assessments of LLMs.
2. Proposes JudgeAgent, a knowledge-aware evaluation framework that dynamically adjusts question difficulty based on the target model’s performance, allowing for fine-grained capability assessment.
3. Provides targeted optimization suggestions by identifying capability gaps, which lead to measurable improvements in model accuracy on follow-up tasks.
4. Conducts extensive experiments and analysis to validate the effectiveness of both the Agent-as-Interviewer paradigm and the JudgeAgent framework across multiple models and datasets.

**Weaknesses:**

1. JudgeAgent’s multi-turn interaction and dynamic question generation introduce significant computational overhead, requiring more resources and time than static evaluation methods.
2. Construction of context graphs, entity extraction, and multi-hop knowledge sampling are resource-intensive and pose scalability challenges, especially in large-scale deployment scenarios.
3. Compared to static benchmarks, dynamic follow-up question generation using LLMs introduces variability, which makes evaluations less reproducible and harder to standardize across repeated runs or different systems.

**Questions:**

1. The study focuses on medical and reasoning tasks. How well does JudgeAgent transfer to other domains?
2. Since the same questions are reused for evaluation after feedback, is there a risk that target models are simply learning the test rather than showing true capability improvement?
3. How can we verify that feedback does not contain the correct answers, and that the model is following the prompt instructions correctly during feedback generation?
4. If we provide the correct answer directly instead of feedback, how much additional performance gain is observed compared to JudgeAgent's feedback-only method?

---

> ### Author Response · Authors · 2025-11-21
> **Responses to Reviewer 7GU8, About the Weaknesses**
>
> Thanks for your comments on the potential weaknesses of our work. Below we respond to these weaknesses point by point.
>
> **1. About time and resources cost:**
>
> Thanks for highlighting the concern about the time and resources cost. We calculate the actual costs of multi-round interactions, as shown in the table below.
>
> | Target     | avg.Time-MedQA | avg.Token-MedQA | avg.Time-MultiHopRAG | avg.Token-MultiHopRAG |
> | ---------- | -------------- | --------------- | -------------------- | --------------------- |
> | GLM4-Flash | 7.24           | 1558            | 23.38                | 2424                  |
> | Qwen3      | 9.95           | 1909            | 30.96                | 3453                  |
>
> Compared to the significant improvement in evaluation effectiveness, these costs are within an acceptable range. Moreover, considering the time required to construct static benchmarks and their risk of data contamination, the overhead of multi-round interactions and dynamic question generation does not exceed the cost of updating static benchmarks.
>
> **2. About scalability:**
>
> Thanks for attention to scalability. We would like respond this question from two aspects.
>
> First, our work focuses on vertical domain application scenarios with a relatively well-defined scope of knowledge. Under the premise of emphasizing domain depth, our approach can construct a context graph from given knowledge sources and, based on this, extend the evaluation coverage as comprehensively as possible within the required knowledge domain to thoroughly evaluate the target model. This is crucial for deploying models in real-world application scenarios.
>
> Second, the algorithms for our graph construction, entity extraction, and knowledge sampling all involve unidirectional traversal, making the complexity acceptable. The extraction of entities from corpora or questions using large language models is a unidirectional process, with each text invoking LLM only once. For knowledge sampling, as described in Sec 3.2, Lines 208-211, the primary time consumption lies in similarity calculations. By precomputing text embeddings and accelerating computations with matrix operations, the time required for this step is comparable to that of invoking LLMs. Additionally, the graph construction process is also unidirectional. When new corpora are added, they are first chunked, and then entities are extracted by traversing the chunks. The chunks are subsequently added to the corresponding entity nodes in the graph, with entity queries and insertions handled via dictionary operations. Under this process, expanding with each new corpus does not require significant time.
>
> **3. About result stability by using LLMs:**
>
> We supplement the experiment results under five different random seeds, calculate the standard deviation, and the results are in the table below:
>
> Results on MedQA:
>
> | Target     | ACC-before   | ACC-after    | CR           | CtE         |
> | ---------- | ------------ | ------------ | ------------ | ----------- |
> | Qwen3      | 91.76 ± 0.09 | 96.18 ± 0.17 | 4.83 ± 0.17  | 0.42 ± 0.08 |
> | GLM4-Flash | 80.36 ± 0.34 | 92.65 ± 0.29 | 13.00 ± 0.38 | 0.72 ± 0.18 |
>
> Results on MultiHop-RAG:
>
> | Target     | ACC-before   | ACC-after    | CR           | CtE         |
> | ---------- | ------------ | ------------ | ------------ | ----------- |
> | Qwen3      | 63.97 ± 0.53 | 70.41 ± 0.59 | 15.78 ± 1.29 | 9.34 ± 1.35 |
> | GLM4-Flash | 51.54 ± 0.38 | 66.56 ± 0.51 | 23.44 ± 0.88 | 8.22 ± 1.03 |
>
> The experiment results indicate that the results across multiple trials are relatively stable, which attests to the controllability of the dynamic evaluation process achieved through context-graph constraints and difficulty control.

---

> ### Author Response · Authors · 2025-11-21
> **Responses to Reviewer 7GU8, About the Questions**
>
> Thanks for your questions. Our responses to these questions are as below:
>
> **1. About the transferability:**
>
> In fact, MultiHopRAG is a dataset from another domain, which is detailed in **Appendix D.1 (Page 14)**. This dataset involves news articles on various topics. In our experiments, we removed its evidence chains to examine the target model's understanding of different domain knowledge in the news. As shown in the main experiment results in **Table 1**, our framework demonstrates domain transferability.
>
> **2. About the risk that the target models simply learned test:**
>
> For this question, we would like to clarify two points.
>
> First, when interacting with the target model via APIs, we did not provide it with its own Q&A history. The Q&A history is solely maintained within the evaluator Agent. Therefore, **the target model cannot memorize the test content during the interaction**.
>
> Second, for whether the target model might have been pre-exposed to the test content, i.e. data contamination, we clarify it from two perspectives. On one hand, the main experiment results in Table 1 and Table 2 show performance improvements, indicating that the evaluation successfully identified deficiencies in the target model. On the other hand, we conducted investigations into data contamination in **Appendix E.1 (Pages 15–16) and E.2 (Pages 16–17)**. In E.1, we examined the evaluation effectiveness on target models that had been contaminated with benchmarks. The results in **Table 5 (Page 16)** demonstrate that even if the target model memorized the test, **its performance on dynamic questions remained poor, confirming that the evaluation can still identify problems**. In E.2, we explored whether the feedback is only effective for questions from the benchmark. The results in **Table 6 (Page 17)** indicate that **the evaluation feedback exhibits a certain degree of generalizability**.
>
> **3. About whether correct answers are contained in feedback:**
>
> For this question, we have demonstrated in **Sec 3.3 Lines 245-246**. During feedback generation, only the question, the target model's response, and whether the response was correct are provided as information. **The correct answer and background knowledge are not included.**
>
> Moreover, we have experimentally validated and analyzed whether the evaluation feedback contains cheating information in **Appendix E.2 (Pages 16–17)**. The results in **Table 6** show that the evaluation effectiveness on similar knowledge questions is comparable to that on seed questions, indicating that **the evaluation feedback does not contain cheating information tailored specifically to the seed questions.**
>
> **4. About experiments that providing correct answers:**
>
> Thank you for your suggestion. We supplemente the comparative experiments with directly providing correct answers, and the results are shown in the table below:
>
> | Target     | ACC-before | ACC-after-Direct | ACC-after-Feedback |
> | ---------- | ---------- | ---------------- | ------------------ |
> | Qwen3      | 91.71      | 99.18            | 96.38              |
> | GLM4-Flash | 80.09      | 98.37            | 92.82              |
> | GPT-4.1    | 84.97      | 98.28            | 92.44              |
>
> As can be observed, when the correct answers are directly provided, nearly all LLMs achieve almost perfect accuracy. This further illustrates the risk of data contamination in existing static benchmarks.

---

> ### Comment · Reviewer_7GU8 · 2025-11-25
>
> Thank you for your response. I will maintain the original rating at this time.

---

> > ### Author Response · Authors · 2025-11-25
> >
> > Thank you for your comments. We are happy to address any other concerns you may have, and we hope our further discussion will enhance your recognition of our work.

---

### Official Review · Reviewer_HDmW · 2025-11-03

**Soundness:** 2
**Presentation:** 3
**Contribution:** 2
**Rating:** 2
**Confidence:** 4

**Summary:**

The paper is motivated by limits of static LLM benchmarks: saturation, data-contamination risks, restricted knowledge scope, and mismatched question difficulty. To address this, the authors propose the agent-as-interviewer method, where an agent conducts multi-turn interviews, invokes knowledge tools to broaden/deepen context for follow-ups, and adapts difficulty to the target model’s current performance. The authors implement JudgeAgent, a three-stage pipeline (benchmark grading, interactive extension, and evaluation feedback). Experiments show that agent-as-interviewer can assess target LLMs effectively and that the evaluation feedback can improve LLM performance.

**Strengths:**

1. This paper is clear and easy to understand.
2. Clear motivation. The paper crisply articulates why static benchmarks fall short (saturation, contamination, scope/difficulty mismatch) and why dynamic evaluation can help.
3. Knowledge-driven dynamic questioning. A context-graph sampling scheme grounds follow-up questions in wider/deeper reference texts from the benchmark knowledge base.

**Weaknesses:**

1. **Result Stability**: Because this work is proposed as an evaluation/benchmarking method, result stability is important. However, there is no empirical stability study across independent runs/seeds, nor any analysis of rank consistency.
2. **Quality & Diversity of Synthesized Questions**: The authors claim that their method can probe the knowledge boundaries of the evaluated LLMs, which presupposes that the synthesized questions are diverse and high-quality. However, there’s no direct assessment of diversity (coverage/novelty/duplication) or question quality (answerability, ambiguity, factuality/hallucination rate).
3. **Judge Reliability**: The framework relies on an LLM-as-judge to score synthesized questions, so the reliability of those judgments is critical. However, the paper does not calibrate the judge against human-labeled ground truth on the synthesized questions.
4. **Data Contamination**: Although dynamic evaluation methods are theoretically less vulnerable to data contamination, the synthesized questions in this work are still derived from the static datasets. Therefore, whether the proposed method can effectively avoid contamination remains unproven and should be validated with controlled experiments.

**Questions:**

1. Please include empirical evidence of result stability across random seeds or independent runs, as well as rank-consistency analysis
2. The work would benefit from quantitative and qualitative evaluation of the synthesized questions, including their answerability, factual accuracy, and diversity.
3. A calibration with human annotations or alternative judges would be beneficial to demonstrate judgment reliability and fairness.

**Details Of Ethics Concerns:**

No ethics concerns.

---

> ### Author Response · Authors · 2025-11-21
> **Responses to Reviewer HDmW**
>
> We sincerely thank reviewer HDmW for valuable comments on our work. Below we respond to the weaknesses and questions.
>
> **1. About the result stability:**
>
> We sincerely thank the reviewer for the attention to the result stability. We supplement experiments with 5 different random seeds, calculate the standard deviation, and the experiment results are shown in the table below:
>
> Results on MedQA:
>
> | Target     | ACC-before   | ACC-after    | CR           | CtE         |
> | ---------- | ------------ | ------------ | ------------ | ----------- |
> | Qwen3      | 91.76 ± 0.09 | 96.18 ± 0.17 | 4.83 ± 0.17  | 0.42 ± 0.08 |
> | GLM4-Flash | 80.36 ± 0.34 | 92.65 ± 0.29 | 13.00 ± 0.38 | 0.72 ± 0.18 |
>
> Results on MultiHop-RAG:
>
> | Target     | ACC-before   | ACC-after    | CR           | CtE         |
> | ---------- | ------------ | ------------ | ------------ | ----------- |
> | Qwen3      | 63.97 ± 0.53 | 70.41 ± 0.59 | 15.78 ± 1.29 | 9.34 ± 1.35 |
> | GLM4-Flash | 51.54 ± 0.38 | 66.56 ± 0.51 | 23.44 ± 0.88 | 8.22 ± 1.03 |
>
> The experiment results demonstrate that the results across multiple random seeds remain stable and are consistent with the ranking of results presented in our paper. This stability benefits from JudgeAgent's context-graph control and knowledge-driven synthesis pathway for evaluation questions. These mechanisms endow the generated questions and the evaluation feedback with controllability and stability.
>
> **2. About the quality and diversity of synthesized questions:**
>
> Thanks for attention to this problem. In fact, compared to previous works, **one of the core contributions of our work is enhancing the factuality and controllability of evaluation questions by constructing a context graph and adopting a knowledge-driven approach to question synthesis**. Previous methods relied entirely on LLMs for question synthesis, which were prone to hallucinations or biases. In contrast, our approach delegates knowledge and question extension to the context graph, ensuring accuracy and relevance through a controlled sampling mechanism (as described in **Sec 3.2, Lines 208-213**). LLMs are only responsible for extracting knowledge entities and generating questions based on reliable knowledge, thereby significantly reducing hallucinations or biases. Additionally, the difficulty adjustment mechanism ensures that questions remain within the normal response range of the target model.
>
> For the diversity, **we have already provided indirect validation through data contamination experiments in Appendix E.1 (Page 15)**. The results in **Table 5** show that contaminated models perform poorly on synthesized questions, which indirectly confirms that synthesized questions differ from seed questions and that their diversity is assured.
>
> **3. About the judge reliability:**
>
> We need to clarify that **our framework does not use LLMs to score synthetic questions**. Considering that the ultimate goal of both evaluation results and feedback is to contribute to the improvement of the target model, and given that neither meta-evaluation with LLMs nor manual validation can guarantee that the validated feedback provides value to the target model, **we instead choose to directly observe the performance improvement of the target model on benchmarks before and after feedback to assess effectiveness. This result is quantifiable, objective, and reproducible.**
>
> **4. About data contamination:**
>
> For the resistance of our framework to data contamination, **we have conducted detailed experiments in Appendix E.1 (Page 15)**, with results presented in **Table 5**. We fine-tuned Llama3-8B-Instruct, Mistral-7B-Instruct-v0.3, and Qwen2.5-7B-Instruct using questions from static benchmarks. By comparing the performance differences between the base models and the fine-tuned models, as well as their performance after evaluation feedback, we validated the resistance of our framework to data contamination.
>
> The results in **Table 5** show that while the fine-tuned models exhibit significant improvement on the original benchmarks, their performance in multi-turn evaluations remains comparable or even slightly decreased, demonstrating that **our framework effectively resists data contamination**.

---

> > ### Author Response · Authors · 2025-11-25
> >
> > Thank you for your valuable comments. We have addressed your points in our rebuttal and hope our responses have fully addressed your concerns.
> > Could you please let us know if any points still need clarification? We look forward to your response.

---

> ### Comment · Reviewer_HDmW · 2025-11-26
>
> Thanks for the authors' responses. Regarding judge reliability, the evaluation pipeline still depends on a LLM to generate both the questions and their reference answers. The current experiments primarily demonstrate the value of the feedback as a training/prompting signal, rather than directly showing that the evaluation scores themselves are reliable. Therefore, I view JudgeAgent more as a data augmentation and feedback generation framework than as an evaluation paradigm that can be directly compared to traditional static benchmarks. Since some of my concerns have been addressed, I will raise the rating to 4.

---

### Official Review · Reviewer_XF41 · 2025-11-05

**Soundness:** 3
**Presentation:** 3
**Contribution:** 2
**Rating:** 6
**Confidence:** 3

**Summary:**

The paper proposes JudgeAgent, an agent-as-interviewer dynamic evaluation framework that conducts multi-turn adaptive interviews with target LLMs. The system integrates 1. benchmark grading, 2. interactive extension，and 3. evaluation feedback generation..
Experiment Results on MedQA, MultiHop-RAG, and QuALITY across multiple LLMs show that JudgeAgent can find weaknesses better than traditional static evaluation and slightly improve accuracy when the model re-answers questions after receiving feedback.

**Strengths:**

- Addresses an important issue beyond static benchmarks to dynamic, adaptive evaluation.
- Context-graph with difficulty control is a novel way of dealing with interactive extension.
- Clear structure with intuitive figures. Easy to follow.
- Empirical improvement across datasets and models with adeqate ablations.

**Weaknesses:**

-The overlaps with prior dynamic / interviewer-based evaluation works makes the novelty limited
- Evaluation quality and feedbacks are evaluated only with LLM accuracy, not expert judgment.
- Context-graph construction, difficulty labeling, and stopping criteria are not clearly explained.
- What is the cost/latency for multi-turn evaluation?
- Limited generalizatio with only QA tasks, what about reasoning or code generation?

**Questions:**

- How is the context graph constructed and difficulty adjusted during interaction?
- What is the cost per evaluation round and trade-off with acc? Latency for multi-turn?
- Could JudgeAgent generalize to other domains (e.g., reasoning, coding)?

---

> ### Author Response · Authors · 2025-11-21
> **Responses to Reviewer XF41**
>
> We sincerely thank reviewer XF41 for the deep and objective comments. Below we respond to the weaknesses and questions.
>
> **1. About the novelty:**
>
> As mentioned in **Figure 1** and **Sec 1, Lines 72-77**, existing works overly rely on the knowledge of the evaluator LLms when interacting with the target model, which is prone to hallucinations. Additionally, they generate follow-up questions solely based on the target's responses, lacking control of difficulty to match the target's actual proficiency. As discussed in **Sec 3 Lines 218-221**, fixed evaluation question difficulty makes it challenging to understand the actual capability boundaries of the target model precisely. These limitations of existing works are also elaborated in **Sec 2.2 Lines 157-160**.
>
> To address these challenges, our work employs a context graph to retrieve relevant knowledge for generating accurate questions. Simultaneously, during the evaluation process, it dynamically adjusts the capability estimation based on the target's performance to control the question's difficulty, ensuring the final evaluation results provide sufficient value for the target model.
>
> **2. About the expert judgement:**
>
> We want to respond to it from two aspects.
>
> First, as shown in **Sec 4.6** and **Appendix H**, manual analysis reveals that the generated questions are highly relevant to the seed knowledge entities, and the feedback points out the target model's deficiency in relative knowledge.
>
> Second, **the ultimate goal of both evaluation results and feedback is to contribute to the improvement of the target model**. Whether using LLMs for meta-evaluation or conducting manual validation, it is difficult to ensure that the validated feedback provides value to the target model. Therefore, we choose to **directly observe the performance improvement of the target model on benchmarks before and after feedback is applied. This result is quantifiable, objective, and reproducible.**
>
> **3. About the context-graph construction, difficulty labeling, and stopping criteria:**
>
> The construction of context graphs has been detailed in **Appendix C (Page 14) and Algorithm 3 (Page 22)**, including how to chunk the orignal texts, how to construct nodes, and how to link nodes.
>
> The difficulty adjusting has been detailed in **Sec 3.1 Line 182-193 and Appendix B (Page 12-13)**, including how we adjust the difficulty based on the target's responses and how we set the score gains and thresholds.
>
> The stopping criterion has been mentioned in **Sec 4, Line 262**. We set a fixed 3 rounds to end the Interactive Extension phase, and the rationale for selecting 3 rounds has been thoroughly analyzed in **Sec 4.5, Parameter Analysis**. As shown in **Figure 3**, there is a significant improvement from round 0 to round 3, while beyond 3 rounds, marginal effects become evident. Considering cost and time, we end the Interactive Extension phase after 3 rounds.
>
> **4. About the cost/latency for multi-turn evaluation:**
>
> Thanks for highlighting the concern about the cost and latency. For the time consumption of multi-round evaluation, we have briefly mentioned it in **Figure 6 (Page 18) of Appendix E.3**. Under the experimental settings of our work, the average time consumption per evaluation question (counted by questions in the benchmark, and the cost per question including consumption across all follow-up stages.) for evaluating GLM4-Flash and Qwen3 is **7.24 s** and **9.95 s**, respectively. More detailed time and token consumption, as shown in the table below:
>
> | Target     | avg.Time-MedQA | avg.Token-MedQA | avg.Time-MultiHopRAG | avg.Token-MultiHopRAG |
> | ---------- | -------------- | --------------- | -------------------- | --------------------- |
> | GLM4-Flash | 7.24           | 1558            | 23.38                | 2424                  |
> | Qwen3      | 9.95           | 1909            | 30.96                | 3453                  |
>
> **5. About the generalization:**
>
> Our work primarily focuses on evaluating the knowledge mastery of LLMs. For instance, our experiments using MedQA and MultiHop-RAG aim to determine whether our framework can help target models optimize their knowledge gaps, while experiments using **QuALITY** investigate whether our framework can **assist target models in enhancing their comprehension and reasoning abilities**. These details have already been provided in **Section 4.1 and Appendix D (Page 14)**.
>
> Code generation tasks are not the central focus of this work. However, our framework holds transferable value. It is possible to adjust the difficulty of coding problems based on factors such as the executability of the code generated by the target model and whether it solves the given problem. Nevertheless, this would require replacing the evaluator Agent's data source with code-related resources and modifying the evaluation approach to incorporate sandbox testing for simulated code execution. We plan to further explore this work in our future works.

---

> > ### Comment · Reviewer_XF41 · 2025-11-22
> >
> > Thanks for your response. I'm clearer on some of the aspects. I tend to keep the original rating.

---

> > > ### Author Response · Authors · 2025-11-25
> > >
> > > Thank you for your feedback on our work. If you have any other concerns, please feel free to discuss them further, and we hope this will enhance your recognition of our work.

---

> > ### Comment · Reviewer_XF41 · 2025-11-26
> >
> > Sorry that was for a different paper. Thanks for your clarification. I have no more questions.

---

### Note · Authors · 2026-01-04

I have read and agree with the venue's withdrawal policy on behalf of myself and my co-authors.